# Hierarchical Generative Modeling for Controllable Speech Synthesis

**Wei-Ning Hsu**[1]* **Yu Zhang**[2] **Ron J. Weiss**[2] **Heiga Zen**[2] **Yonghui Wu**[2] **Yuxuan Wang**[2]
**Yuan Cao**[2] **Ye Jia**[2] **Zhifeng Chen**[2] **Jonathan Shen**[2] **Patrick Nguyen**[2] **Ruoming Pang**[2]
[1]Massachusetts Institute of Technology     [2]Google Inc.
wnhsu@csail.mit.edu, {ngyuzh,ronw}@google.com

## Abstract

This paper proposes a neural sequence-to-sequence text-to-speech (TTS) model which can control latent attributes in the generated speech that are rarely annotated in the training data, such as speaking style, accent, background noise, and recording conditions. The model is formulated as a conditional generative model based on the variational autoencoder (VAE) framework, with two levels of hierarchical latent variables. The first level is a categorical variable, which represents attribute groups (e.g. clean/noisy) and provides interpretability. The second level, conditioned on the first, is a multivariate Gaussian variable, which characterizes specific attribute configurations (e.g. noise level, speaking rate) and enables disentangled fine-grained control over these attributes. This amounts to using a Gaussian mixture model (GMM) for the latent distribution. Extensive evaluation demonstrates its ability to control the aforementioned attributes. In particular, we train a high-quality controllable TTS model on real found data, which is capable of inferring speaker and style attributes from a noisy utterance and use it to synthesize *clean* speech with controllable speaking style.

## 1 Introduction

Recent development of neural sequence-to-sequence TTS models has shown promising results in generating high fidelity speech without the need of handcrafted linguistic features (Sotelo et al., 2017; Wang et al., 2017; Arık et al., 2017; Shen et al., 2018). These models rely heavily on a encoder-decoder neural network structure (Sutskever et al., 2014; Bahdanau et al., 2015) that maps a text sequence to a sequence of speech frames. Extensions to these models have shown that attributes such as speaker identity can be controlled by conditioning the decoder on additional attribute labels (Arik et al., 2017; 2018; Jia et al., 2018).

There are many speech attributes aside from speaker identity that are difficult to annotate, such as speaking style, prosody, recording channel, and noise levels. Skerry-Ryan et al. (2018); Wang et al. (2018) model such *latent* attributes through conditional auto-encoding, by extending the decoder inputs to include a vector inferred from the target speech which aims to capture the residual attributes that are not specified by other input streams, in addition to text and a speaker label. These models have shown convincing results in synthesizing speech that resembles the prosody or the noise conditions of the reference speech, which may not have the same text or speaker identity as the target speech.

Nevertheless, the presence of multiple latent attributes is common in crowdsourced data such as (Panayotov et al., 2015), in which prosody, speaker, and noise conditions all vary simultaneously. Using such data, simply copying the latent attributes from a reference is insufficient if one desires to synthesize speech that mimics the prosody of the reference, but is in the same noise condition as another. If the latent representation were *disentangled*, these generating factors could be controlled independently. Furthermore, it is can useful to construct a systematic method for synthesizing speech with random latent attributes, which would facilitate data augmentation (Tjandra et al., 2017; 2018; Hsu et al., 2017b; 2018; Hayashi et al., 2018) by generating diverse examples. These properties were not explicitly addressed in the previous studies, which model variation of a single latent attribute.

---

*Work performed while interning at Google Brain.

Motivated by the applications of sampling, inferring, and independently controlling individual attributes, we build off of Skerry-Ryan et al. (2018) and extend Tacotron 2 (Shen et al., 2018) to model two separate latent spaces: one for labeled (i.e. related to speaker identity) and another for unlabeled attributes. Each latent variable is modeled in a variational autoencoding (Kingma & Welling, 2014) framework using Gaussian mixture priors. The resulting latent spaces (1) learn disentangled attribute representations, where each dimension controls a different generating factor; (2) discover a set of interpretable clusters, each of which corresponds to a representative mode in the training data (e.g., one cluster for clean speech and another for noisy speech); and (3) provide a systematic sampling mechanism from the learned prior. The proposed model is extensively evaluated on four datasets with subjective and objective quantitative metrics, as well as comprehensive qualitative studies. Experiments confirm that the proposed model is capable of controlling speaker, noise, and style independently, even when variation of all attributes is present but unannotated in the train set.

Our main contributions are as follows:

- We propose a principled probabilistic hierarchical generative model, which improves (1) sampling stability and disentangled attribute control compared to e.g. the GST model of Wang et al. (2018), and (2) interpretability and quality compared to e.g. Akuzawa et al. (2018).

- The model formulation explicitly factors the latent encoding by using two mixture distributions to separately model supervised speaker attributes and latent attributes in a disentangled fashion. This makes it straightforward to condition the model output on speaker and latent encodings inferred from different reference utterances.

- To the best of our knowledge, this work is the first to train a high-quality controllable text-to-speech system on real found data containing significant variation in recording condition, speaker identity, as well as prosody and style. Previous results on similar data focused on speaker modeling (Ping et al., 2018; Nachmani et al., 2018; Arik et al., 2018; Jia et al., 2018), and did not explicitly address modeling of prosody and background noise. Leveraging disentangled speaker and latent attribute encodings, the proposed model is capable of inferring the speaker attribute representation from a noisy utterance spoken by a previously unseen speaker, and using it to synthesize high-quality clean speech that approximates the voice of that speaker.

## 2 MODEL

Tacotron-like TTS systems take a text sequence $\mathbf{Y}_t$ and an optional observed categorical label (e.g. speaker identity) $\mathbf{y}_o$ as input, and use an autoregressive decoder to predict a sequence of acoustic features $\mathbf{X}$ frame by frame. Training such a system to minimize a mean squared error reconstruction loss can be regarded as fitting a probabilistic model $p(\mathbf{X} \mid \mathbf{Y}_t, \mathbf{y}_o) = \prod_n p(\mathbf{x}_n \mid \mathbf{x}_1, \mathbf{x}_2, \ldots, \mathbf{x}_{n-1}, \mathbf{Y}_t, \mathbf{y}_o)$ that maximizes the likelihood of generating the training data, where the conditional distribution of each frame $\mathbf{x}_n$ is modeled as fixed-variance isotropic Gaussian whose mean is predicted by the decoder at step $n$. Such a model effectively integrates out other unlabeled latent attributes like prosody, and produces a conditional distribution with higher variance. As a result, the model would opaquely produce speech with unpredictable latent attributes. To enable control of those attributes, we adopt a graphical model with hierarchical latent variables, which captures such attributes. Below we explain how the formulation leads to interpretability and disentanglement, supports sampling, and propose efficient inference and training methods.

### 2.1 CONDITIONAL GENERATIVE MODEL WITH HIERARCHICAL LATENT VARIABLES

Two latent variables $\mathbf{y}_l$ and $\mathbf{z}_l$ are introduced in addition to the observed variables, $\mathbf{X}$, $\mathbf{Y}_t$, and $\mathbf{y}_o$, as shown in the graphical model in the left of Figure 1. $\mathbf{y}_l$ is a $K$-way categorical discrete variable, named *latent attribute class*, and $\mathbf{z}_l$ is a $D$-dimensional continuous variable, named *latent attribute representation*. Throughout the paper, we use $\boldsymbol{y}_*$ and $\boldsymbol{z}_*$ to denote discrete and continuous variables, respectively. To generate speech $\boldsymbol{X}$ conditioned on the text $\boldsymbol{Y}_t$ and observed attribute $\boldsymbol{y}_o$, $\mathbf{y}_l$ is first sampled from its prior, $p(\mathbf{y}_l)$, then a latent attribute representation $\mathbf{z}_l$ is sampled from the conditional distribution $p(\mathbf{z}_l \mid \mathbf{y}_l)$. Finally, a sequence of speech frames is drawn from $p(\mathbf{X} \mid \mathbf{Y}_t, \boldsymbol{y}_o, \boldsymbol{z}_l)$, parameterized by the synthesizer neural network. The joint probability can be written as:

$$p(\mathbf{X}, \mathbf{y}_l, \mathbf{z}_l \mid \mathbf{Y}_t, \mathbf{y}_o) = p(\mathbf{X} \mid \mathbf{Y}_t, \mathbf{y}_o, \mathbf{z}_l)\, p(\mathbf{z}_l \mid \mathbf{y}_l)\, p(\mathbf{y}_l). \tag{1}$$

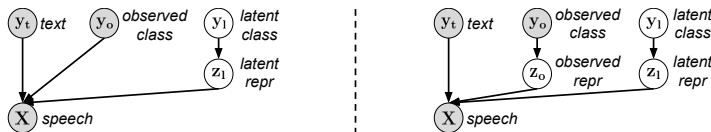

Figure 1: Graphical model representation of the proposed models. *Observed class* often corresponds to the speaker label. The left illustrates equation 1, and the right illustrates the extension from Section 2.3. The grey and white nodes correspond to observed and latent variables.

Specifically, it is assumed that $p(\mathbf{y}_l) = K^{-1}$ to be a non-informative prior to encourage every component to be used, and $p(\mathbf{z}_l \mid \mathbf{y}_l) = \mathcal{N}(\boldsymbol{\mu}_{\mathbf{y}_l}, \mathrm{diag}(\boldsymbol{\sigma}_{\mathbf{y}_l}))$ to be diagonal-covariance Gaussian with learnable means and variances. As a result, the marginal prior of $\mathbf{z}_l$ becomes a GMM with diagonal covariances and equal mixture weights. We hope this GMM latent model can better capture the complexity of unseen attributes. Furthermore, in the presence of natural clusters of unseen attributes, the proposed model can achieve interpretability by learning to assign instances from different clusters to different mixture components. The covariance matrix of each mixture component is constrained to be diagonal to encourage each dimension to capture a statistically uncorrelated factor.

## 2.2 VARIATIONAL INFERENCE AND TRAINING

The conditional output distribution $p(\mathbf{X} \mid \mathbf{Y}_t, \mathbf{y}_o, \mathbf{z}_l)$ is parameterized with a neural network. Following the VAE framework of Kingma & Welling (2014), a variational distribution $q(\mathbf{y}_l \mid \mathbf{X})\, q(\mathbf{z}_l \mid \mathbf{X})$ is used to approximate the posterior $p(\mathbf{y}_l, \mathbf{z}_l \mid \mathbf{X}, \mathbf{Y}_t, \mathbf{y}_o)$, which assumes that the posterior of unseen attributes is independent of the text and observed attributes. The approximated posterior for $\mathbf{z}_l$, $q(\mathbf{z}_l \mid \mathbf{X})$, is modeled as a Gaussian distribution with diagonal covariance matrix, whose mean and variance are parameterized by a neural network. For $q(\mathbf{y}_l \mid \mathbf{X})$, instead of introducing another neural network, we configure it to be an approximation of $p(\mathbf{y}_l \mid \mathbf{X})$ that reuses $q(\mathbf{z}_l \mid \mathbf{X})$ as follows:

$$p(\mathbf{y}_l|\mathbf{X}) = \int_{\mathbf{z}_l} p(\mathbf{y}_l \mid \mathbf{z}_l)\, p(\mathbf{z}_l|\mathbf{X})\, d\mathbf{z}_l = \mathbb{E}_{p(\mathbf{z}_l|\mathbf{X})}\left[p(\mathbf{y}_l \mid \mathbf{z}_l)\right] \approx \mathbb{E}_{q(\mathbf{z}_l|\mathbf{X})}\left[p(\mathbf{y}_l \mid \mathbf{z}_l)\right] := q(\mathbf{y}_l|\mathbf{X}) \quad (2)$$

which enjoys the closed-form solution of Gaussian mixture posteriors, $p(\mathbf{y}_l \mid \mathbf{z}_l)$. Similar to VAE, the model is trained by maximizing its evidence lower bound (ELBO), as follows:

$$\mathcal{L}(p, q; \mathbf{X}, \mathbf{Y}_t, \mathbf{y}_o) = \mathbb{E}_{q(\mathbf{z}_l|\mathbf{X})}[\log p(\mathbf{X} \mid \mathbf{Y}_t, \mathbf{y}_o, \mathbf{z}_l)]$$
$$- \mathbb{E}_{q(\mathbf{y}_l|\mathbf{X})}[D_{KL}(q(\mathbf{z}_l \mid \mathbf{X}) \,\|\, p(\mathbf{z}_l \mid \mathbf{y}_l))] - D_{KL}(q(\mathbf{y}_l \mid \mathbf{X}) \,\|\, p(\mathbf{y}_l)) \quad (3)$$

where $q(\mathbf{z}_l \mid \mathbf{X})$ is estimated via Monte Carlo sampling, and all components are differentiable thanks to reparameterization. Details can be found in Appendix A.

## 2.3 A CONTINUOUS ATTRIBUTE SPACE FOR CATEGORICAL OBSERVED LABELS

Categorical observed labels, such as speaker identity, can often be seen as a categorization from a continuous attribute space, which for example could model a speaker's characteristic $F_0$ range and vocal tract shape. Given an observed label, there may still be some variation of these attributes. We are interested in learning this continuous attribute space for modeling within-class variation and inferring a representation from an instance of an unseen class for one-shot learning.

To achieve this, a continuous latent variable, $\mathbf{z}_o$, named the *observed attribute representation*, is introduced between the categorical observed label $\mathbf{y}_o$ and speech $\mathbf{X}$, as shown on the right of Figure 1. Each observed class (e.g. each speaker) forms a mixture component in this continuous space, whose conditional distribution is a diagonal-covariance Gaussian $p(\mathbf{z}_o \mid \mathbf{y}_o) = \mathcal{N}(\boldsymbol{\mu}_{\mathbf{y}_o}, \mathrm{diag}(\boldsymbol{\sigma}_{\mathbf{y}_o}))$. With this formulation, speech from an observed class $\mathbf{y}_o$ is now generated by conditioning on $\mathbf{Y}_t, \mathbf{z}_l$, and a sample $\mathbf{z}_o$ drawn from $p(\mathbf{z}_o \mid \mathbf{y}_o)$. As before, a variational distribution $q(\mathbf{z}_o \mid \mathbf{X})$, parameterized by a neural network, is used to approximate the true posterior, where the ELBO becomes:

$$\mathcal{L}_o(p, q; \mathbf{X}, \mathbf{Y}_t, \mathbf{y}_o) = \mathbb{E}_{q(\mathbf{z}_o|\mathbf{X})q(\mathbf{z}_l|\mathbf{X})}[\log p(\mathbf{X} \mid \mathbf{Y}_t, \mathbf{z}_o, \mathbf{z}_l)] - D_{KL}(q(\mathbf{z}_o \mid \mathbf{X}) \,\|\, p(\mathbf{z}_o \mid \mathbf{y}_o))$$
$$- \mathbb{E}_{q(\mathbf{y}_l|\mathbf{X})}[D_{KL}(q(\mathbf{z}_l \mid \mathbf{X}) \,\|\, p(\mathbf{z}_l \mid \mathbf{y}_l))] - D_{KL}(q(\mathbf{y}_l \mid \mathbf{X}) \,\|\, p(\mathbf{y}_l)). \quad (4)$$

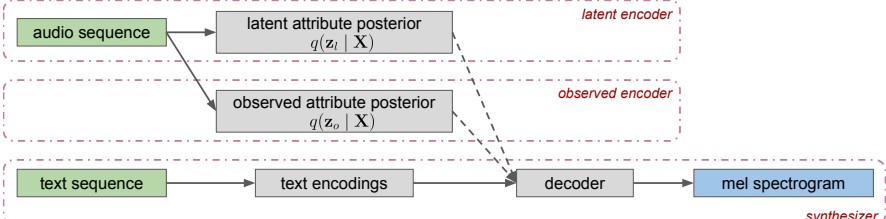

Figure 2: Training configuration of the GMVAE-Tacotron model. Dashed lines denotes sampling. The model is comprised of three modules: a synthesizer, a latent encoder, and an observed encoder.

To encourage $\mathbf{z}_o$ to disentangle observed attributes from latent attributes, the variances of $p(\mathbf{z}_o \mid \mathbf{y}_o)$ are initialized to be smaller than those of $p(\mathbf{z}_l \mid \mathbf{y}_l)$. The intuition is that this space should capture variation of attributes that are highly correlated with the observed labels, so the conditional distribution of all dimensions should have relatively small variance for each mixture component. Experimental results verify the effectiveness, and similar design is used in Hsu et al. (2017a). In the extreme case where the variance is fixed and approaches zero, this formulation converges to using an lookup table.

## 2.4 NEURAL NETWORK ARCHITECTURE

We parameterize three distributions: $p(\mathbf{X}|\mathbf{Y}_t, \mathbf{z}_o, \mathbf{z}_l)$ (or $p(\mathbf{X} \mid \mathbf{Y}_t, \mathbf{z}_o, \mathbf{z}_l)$), $q(\mathbf{z}_o|\mathbf{X})$, and $q(\mathbf{z}_l|\mathbf{X})$ with neural networks, referred to in Figure 2 as the *synthesizer*, *observed encoder*, and *latent encoder*, respectively. The synthesizer is based on the Tacotron 2 architecture (Shen et al., 2018), which consists of a text encoder and an autoregressive speech decoder. The former maps $\mathbf{Y}_t$ to a sequence of text encodings $\mathbf{Z}_t$, and the latter predicts the mean of $p(\mathbf{x}_n \mid \mathbf{x}_1, \ldots, \mathbf{x}_{n-1}, \mathbf{Z}_t, \mathbf{y}_o, \mathbf{z}_l)$ at each step $n$. We inject the latent variables $\mathbf{z}_l$ and $\mathbf{y}_o$ (or $\mathbf{z}_o$) into the decoder by concatenating them to the decoder input at each step. Text $\mathbf{Y}_t$ and speech $\mathbf{X}$ are represented as a sequence of phonemes and a sequence of mel-scale filterbank coefficients, respectively. To speed up inference, we use a WaveRNN-based neural vocoder (Kalchbrenner et al., 2018) instead of WaveNet (van den Oord et al., 2016), to invert the predicted mel-spectrogram to a time-domain waveform.

The two posteriors, $q(\mathbf{z}_l \mid \mathbf{X})$ and $q(\mathbf{z}_o \mid \mathbf{X})$, are both parameterized by a recurrent encoder that maps a variable-length mel-spectrogram to two fixed-dimensional vectors, corresponding to the posterior mean and log variance, respectively. Full architecture details can be found in Appendix B.

## 3 RELATED WORK

The proposed GMVAE-Tacotron model is most related to Skerry-Ryan et al. (2018), Wang et al. (2018), Henter et al. (2018), which introduce a reference embedding to model prosody or noise. The first uses an autoencoder to extract a prosody embedding from a reference speech spectrogram. The second Global Style Token (GST) model constrains a reference embedding to be a weighted combination of a fixed set of learned vectors, while the third further restricts the weights to be one-hot, and is built on a conventional parametric speech synthesizer (Zen et al., 2009). The main focus of these approaches was style transfer from a reference audio example. They provide neither a systematic sampling mechanism nor disentangled representations as we show in Section 4.3.1.

Similar to these approaches, Akuzawa et al. (2018) extend VoiceLoop (Taigman et al., 2018) with a latent reference embedding generated by a VAE, using a centered fixed-variance isotropic Gaussian prior for the latent attributes. This provides a principled mechanism for sampling from the latent distribution, but does not provide interpretability. In contrast, GMVAE-Tacotron models latent attributes using a mixture distribution, which allows automatic discovery of latent attribute clusters. This structure makes it easier to interpret the underlying latent space. Specifically, we show in Section 4.1 that the mixture parameters can be analyzed to understand what each component corresponds to, similar to GST. In addition, the most distinctive dimensions of the latent space can be identified using an inter-/intra-component variance ratio, which e.g. can identify the dimension controlling the background noise level as shown in Section 4.2.2.

Finally, the extension described in Section 2.3 adds a second mixture distribution to additionally models speaker attributes. This formulation learns disentangled speaker and latent attribute represen-

tations, which can be used to approximate the voice of speakers previously unseen during training. This speaker model is related to Arik et al. (2018), which controls the output speaker identity using speaker embeddings, and trains a separate regression model to predict them from the audio. This can be regarded as a special case of the proposed model where the variance of $\mathbf{z}_o$ is set to be almost zero, such that a speaker always generates a fixed representation; meanwhile, the posterior model $q(\mathbf{z}_o \mid \mathbf{X})$ corresponds to their embedding predictor, because it now aims to predict a fixed embedding for each speaker.

Using a mixture distribution for latent variables in a VAE was explored in Dilokthanakul et al. (2016); Nalisnick et al. (2016), and Jiang et al. (2017) for unconditional image generation and text topic modeling. These models correspond to the sub-graph $\mathbf{y}_l \rightarrow \mathbf{z}_l \rightarrow \mathbf{X}$ in Figure 1. The proposed model provides extra flexibility to model both latent and observed attributes in a conditional generation scenario. Hsu et al. (2017a) similarly learned disentangled representations at the variable level (i.e. disentangling $\mathbf{z}_l$ and $\mathbf{z}_o$) by defining different priors for different latent variables. Higgins et al. (2017) also used a prior with diagonal covariance matrix to disentangle different embedding dimensions. Our model provides additional flexibility by learning a different variance in each mixture component.

## 4 EXPERIMENTS

The proposed GMVAE-Tacotron was evaluated on four datasets, spanning a wide degree of variations in speaker, recording channel conditions, background noise, prosody, and speaking styles. For all experiments, $\mathbf{y}_o$ was an observed categorical variable whose cardinality is the number of speakers in the training set if used, $\mathbf{y}_l$ was configured to be a 10-way categorical variable ($K = 10$), and $\mathbf{z}_l$ and $\mathbf{z}_o$ (if used) were configured to be 16-dimensional variables ($D = 16$). Tacotron 2 (Shen et al., 2018) with a speaker embedding table was used as the baseline for all experiments. For all other variants (e.g., GST), the reference encoder follows Wang et al. (2018). Each model was trained for at least 200k steps to maximize the ELBO in equation 3 or equation 4 using the Adam optimizer. A list of detailed hyperparameter settings can be found in Appendix C. Quantitative subjective evaluations relied on crowd-sourced mean opinion scores (MOS) rating the naturalness of the synthesized speech by native speakers using headphones, with scores ranging from 1 to 5 in increments of 0.5. For single speaker datasets each sample was rated by 6 raters, while for other datasets each sample was rated by a single rater. We strongly encourage readers to listen to the samples on the demo page.[1]

### 4.1 MULTI-SPEAKER ENGLISH CORPUS

To evaluate the ability of GMVAE-Tacotron to model speaker variation and discover meaningful speaker clusters, we used a proprietary dataset of 385 hours of high-quality English speech from 84 professional voice talents with accents from the United States (US), Great Britain (GB), Australia (AU), and Singapore (SG). Speaker labels were not seen during training ($\mathbf{y}_o$ and $\mathbf{z}_o$ were unused), and were only used for evaluation.

To probe the interpretability of the model, we computed the distribution of mixture components $\mathbf{y}_l$ for utterances of a particular accent or gender. Specifically, we collected at most 100 utterances from each of the 44 speakers with at least 20 test utterances (2,332 in total), and assigned each utterance to the component with the highest posterior probability: $\arg\max_{\mathbf{y}_l} q(\mathbf{y}_l | \mathbf{X})$.

Figure 3 plots the assignment distributions for each gender and accent in this set. Most components were only used to model speakers from one gender. Each component which modeled both genders (0, 2, and 9) only represented a subset of accents (US, US, and AU/GB, respectively). We also found that the several components which modeled US female speakers (3, 5, and 6) actually modeled groups of speakers with distinct characteristics, e.g. different $F_0$ ranges as shown in Appendix E. To quantify the association between speaker and mixture components, we computed the assignment consistency w.r.t. speaker:

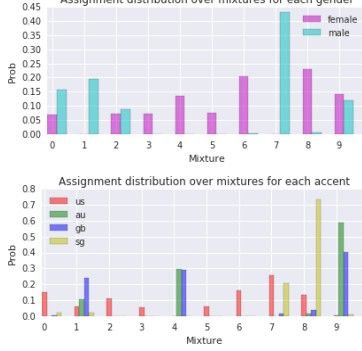

Figure 3: Assignment distribution over $\mathbf{y}_l$ for each gender (upper) and for each accent (lower).

---

[1]`https://google.github.io/tacotron/publications/gmvae_controllable_tts`

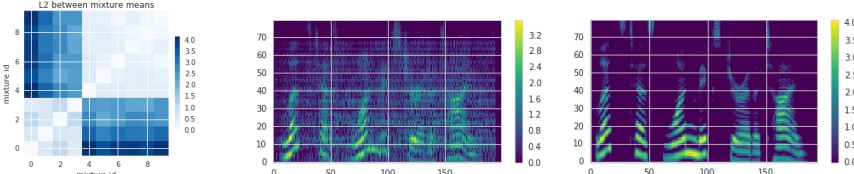

Figure 4: Left: Euclidean distance between the means of each mixture component pair. Right: Decoding the same text conditioned on the mean of a noisy (center) and a clean component (right).

$\frac{1}{M} \sum_{i=1}^{N} \sum_{j=1}^{N_i} \mathbb{1}_{y_{ij}=\hat{y}_i}$ where $M$ is the number of utterances, $y_{ij}$ is the component assignment of utterance $j$ from speaker $i$, and $\hat{y}_i$ is the mode of $\{y_{ij}\}_{j=1}^{N_i}$. The resulting consistency was 92.9%, suggesting that the components group utterances by speaker and group speakers by gender or accent.

We also explored what each dimension of $\mathbf{z}_l$ controlled by decoding with different values of the target dimension, keeping all other factors fixed. We discovered that there were individual dimensions which controlled $F_0$, speaking rate, accent, length of starting silence, etc., demonstrating the disentangled nature of the learned latent attribute representation. Appendix E contains visualization of attribute control and additional quantitative evaluation of using $\mathbf{z}_l$ for gender/accent/speaker classification.

## 4.2 NOISY MULTI-SPEAKER ENGLISH CORPUS

High quality data can be both expensive and time consuming to record. Vast amounts of rich real-life expressive speech are often noisy and difficult to label. In this section we demonstrate that our model can synthesize clean speech directly from noisy data by disentangling the background noise level from other attributes, allowing it to be controlled independently. As a first experiment, we artificially generated training sets using a room simulator (Kim et al., 2017) to add background noise and reverberation to clean speech from the multi-speaker English corpus used in the previous section. We used music and ambient noise sampled from YouTube and recordings of "daily life" environments as noise signals, mixed at signal-to-noise ratios (SNRs) ranging from 5–25dB. The reverberation time varied between 100 and 900ms. Noise was added to a random selection of 50% of utterances by each speaker, holding out two speakers (one male and one female) for whom noise was added to all of their utterances. This construction was used to evaluate the ability of the model to synthesize clean speech for speakers whose training utterances were all corrupted by noise. In this experiment, we provided speaker labels $\mathbf{y}_o$ as input to the decoder, and only expect the latent attribute representations $\mathbf{z}_l$ to capture the acoustic condition of each utterance.

### 4.2.1 IDENTIFYING MIXTURE COMPONENTS THAT GENERATE CLEAN/NOISY SPEECH

Unlike clustering speakers, we expected that latent attributes would naturally divide into two categories: clean and noisy. To verify this hypothesis, we plotted the Euclidean distance between means of each pair of components on the left of Figure 4, which clearly form two distinct clusters. The right two plots in Figure 4 show the mel-spectrograms of two synthesized utterances of the same text and speaker, conditioned on the means of two different components, one from each group. It clearly presents the samples (in fact, all the samples) drawn from components in group one were noisy, while the samples drawn from the other components were clean. See Appendix F for more examples.

### 4.2.2 CONTROL OF THE BACKGROUND NOISE LEVEL

We next explored if the level of noise was dominated by a single latent dimension, and whether we could determine such a dimension automatically. For this purpose, we adopted a per-dimension LDA, which computed a between and within-mixture scattering ratio: $r_d = \sum_{\mathbf{y}_l=1}^{K} p(\mathbf{y}_l)(\boldsymbol{\mu}_{\mathbf{y}_l,d} - \bar{\boldsymbol{\mu}}_{l,d})^2 \ / \ \sum_{\mathbf{y}_l=1}^{K} p(\mathbf{y}_l)\boldsymbol{\sigma}_{\mathbf{y}_l,d}^2$, where $\boldsymbol{\mu}_{\mathbf{y}_l,d}$ and $\boldsymbol{\sigma}_{\mathbf{y}_l,d}$ are the $d$-th dimension mean and variance of mixture component $\mathbf{y}_l$, and $\bar{\boldsymbol{\mu}}_{l,d}$ is the $d$-th dimension mean of the marginal distribution $p(\mathbf{z}_l) = \sum_{\mathbf{y}_l=1}^{K} p(\mathbf{y}_l) p(\mathbf{z}_l \mid \mathbf{y}_l)$. This is a scale-invariant metric of the degree of separation between components in each latent dimension.

We discovered that the most discriminative dimension had a scattering ratio $r_{13} = 21.5$, far larger than the second largest $r_{11} = 0.6$. Drawing samples and traversing values along the target dimension

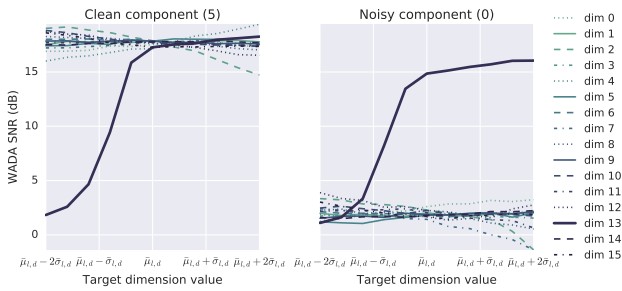

Figure 5: SNR as a function of the value in each latent dimension, comparing clean (left) and noisy (right) components.

Table 1: MOS and SNR comparison among clean original audio, baseline, GST, VAE, and GMVAE models.

| Model | MOS | SNR |
|---|---|---|
| Original | $4.48 \pm 0.04$ | 17.71 |
| Baseline | $2.87 \pm 0.25$ | 11.56 |
| GST | $3.32 \pm 0.13$ | 14.43 |
| VAE | $3.55 \pm 0.17$ | 12.91 |
| GMVAE | $\mathbf{4.25 \pm 0.13}$ | **17.20** |

while keeping others fixed demonstrates that dimension's effect on the output. To determine the effective range of the target dimension, we approximate the multimodal distribution as a Gaussian and evaluate values spanning four standard deviations $\bar{\sigma}_{l,d}$ around the mean. To quantify the effect on noise level, we estimate the SNR without a reference clean signal following Kim & Stern (2008).

Figure 5 plots the average estimated SNR over 200 utterances from two speakers as a function of the value in each latent dimension, where values for other dimensions are the mean of a clean component (left) or that of a noisy component (right). The results show that the noise level was clearly controlled by manipulating the 13th dimension, and remains nearly constant as the other dimensions vary, verifying that control has been isolated to the identified dimension. The small degree of variation, e.g. in dimensions 2 and 4, occurs because some of those dimensions control attributes which directly affect the synthesized noise, such as type of noise (musical/white noise) and initial background noise offset, and therefore also affect the estimated SNR. Appendix F.2 contains an additional spectrogram demonstration of noise level control by manipulating the identified dimension.

### 4.2.3 SYNTHESIZING CLEAN SPEECH FOR NOISY SPEAKERS

In this section, we evaluated synthesis quality for the two held out noisy speakers. Evaluation metrics included subjective naturalness MOS ratings and an objective SNR metric. Table 1 compares the proposed model with a baseline, a 16-token GST, and a VAE variant which replaces the GMM prior with an isotropic Gaussian. To encourage synthesis of clean audio under each model we manually selected the cleanest token (weight=0.15) for GST, used the Gaussian prior mean (i.e. a zero vector) for VAE, and the mean of a clean component for GMVAE. For the VAE model, the mean captured the average condition, which still exhibited a moderate level of noise, resulting in a lower SNR and MOS. The generated speech from the GST was cleaner, however raters sometimes found its prosody to be unnatural. Note that it is possible that another token would obtain a different trade-off between prosody and SNR, and using multiple tokens could improve both. Finally, the proposed model synthesized both natural and high-quality speech, with the highest MOS and SNR.

### 4.3 SINGLE-SPEAKER AUDIOBOOK CORPUS

Prosody and speaking style is another important factor for human speech other than speaker and noise. Control of these aspects of the synthesize speech is essential to building an expressive TTS system. In this section, we evaluated the ability of the proposed model to sample and control speaking styles. A single speaker US English audiobook dataset of 147 hours, recorded by professional speaker, Catherine Byers, from the 2013 Blizzard Challenge (King & Karaiskos, 2013) is used for training. The data incorporated a wide range of prosody variation. We used an evaluation set of 150 audiobook sentences, including many long phrases. Table 2 shows the naturalness MOS between baseline and proposed model conditioning on the same $\mathbf{z}_l$, set to the mean of a selected $\mathbf{y}_l$, for all utterances. The results show that the prior already captured a common prosody, which could be used to synthesize more naturally sounding speech with a lower variance compared to the baseline.

Table 2: MOS comparison of the original audio, baseline and GMVAE.

| Model | MOS |
|---|---|
| Original | $4.67 \pm 0.04$ |
| Baseline | $4.29 \pm 0.11$ |
| Proposed | $\mathbf{4.67 \pm 0.07}$ |

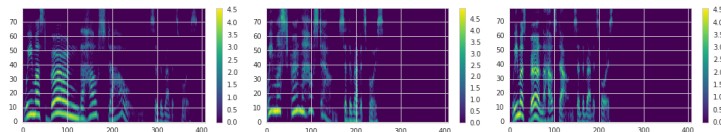

Figure 6: Mel-spectrograms of three samples with the same text, *"We must burn the house down! said the Rabbit's voice."* drawn from the proposed model, showing variation in speed, $F_0$, and pause duration.

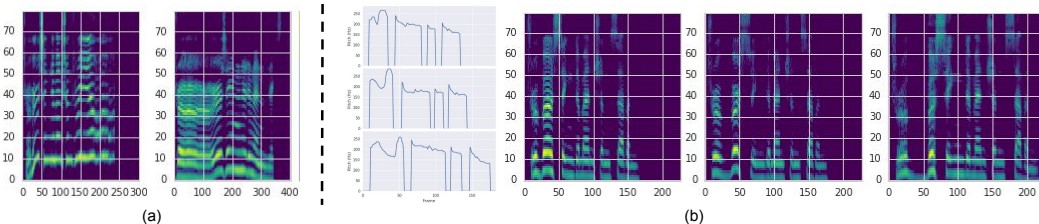

(a)                                   (b)

Figure 7: (a) Mel-spectrograms of two unnatural GST samples when setting the weight for one token -0.1: first with tremolo at the end, and second with abnormally long duration for the first syllable. (b) $F_0$ tracks and spectrograms from GMVAE-Tacotron using different values for the "speed" dimension.

### 4.3.1 STYLE SAMPLING AND DISENTANGLED CONTROL

Compared to GST, one primary advantage of the proposed model is that it supports random sampling of natural speech from the prior. Figure 6 illustrates such samples, where the same text is synthesized with wide variation in speaking rate, rhythm, and $F_0$. In contrast, the GST model does not define a prior for normalized token weights, requiring weights to be chosen heuristically or by fitting a distribution after training. Empirically we found that the GST weight simplex was not fully exploited during training and that careful tuning was required to find a stable sampling region.

An additional advantage of GMVAE-Tacotron is that it learns a representation which disentangles these attributes, enabling them to be controlled independently. Specifically, latent dimensions in the proposed model are conditionally independent, while token weights of GST are in fact correlated. Figure 7(b) contains an example of the proposed model traversing the "speed" dimension with three values: $\bar{\boldsymbol{\mu}}_{l,d} - 2\bar{\boldsymbol{\sigma}}_{l,d}$, $\bar{\boldsymbol{\mu}}_{l,d}$, $\bar{\boldsymbol{\mu}}_{l,d} + 2\bar{\boldsymbol{\sigma}}_{l,d}$, plotted accordingly from left to right, where $\bar{\boldsymbol{\mu}}_{l,d}$ and $\bar{\boldsymbol{\sigma}}_{l,d}$ are the marginal distribution mean and standard deviation, respectively, of that dimension. Their $F_0$ tracks, obtained using the YIN (De Cheveigné & Kawahara, 2002) $F_0$ tracker, are shown on the left. From these we can observe that the shape of the $F_0$ contours did not change much. They were simply stretched horizontally, indicating that only the speed was manipulated. In contrast, the style control of GST is more entangled, as shown in Wang et al. (2018, Figure 3(a)), where the $F_0$ also changed while controlling speed. Appendix G contains a quantitative analysis of disentangled latent attribute control, and additional evaluation of style transfer, demonstrating the ability of the proposed the model to synthesize speech that resembles the prosody of a reference utterance.

### 4.4 CROWD-SOURCED AUDIOBOOK CORPUS

We used an audiobook dataset[2] derived from the same subset of LibriVox audiobooks used for the LibriSpeech corpus (Panayotov et al., 2015), but sampled at 24kHz and segmented differently, making it appropriate for TTS instead of speech recognition. The corpus contains recordings from thousands of speakers, with wide variation in recording conditions and speaking style. Speaker identity is often highly correlated with the recording channel and background noise level, since many speakers tended to use the same microphone in a consistent recording environment. The ability to disentangle and control these attributes independently is essential to synthesizing high-quality speech for all speakers.

---

[2]This dataset will be open-sourced soon.

Table 3: SNR of original audio, baseline, and the proposed models with different conditioned $z_l$, on different speakers.

| Set | Original | Baseline | Proposed | | |
|-----|----------|----------|----------|--------|-----------|
| | | | mean | latent | latent-dn |
| SC | 18.61 | 14.33 | 15.90 | 16.28 | **17.94** |
| SN | 11.80 | 9.69 | 15.82 | 6.78 | **18.94** |
| UC | 20.39 | N/A | 15.70 | 16.40 | **18.83** |
| UN | 10.92 | N/A | 15.27 | 4.81 | **16.89** |

Table 4: Subjective preference (%) between baseline and proposed model with denoised $z_l$ on the set of "seen noisy" (SN) speakers.

| Baseline | Neutral | Proposed |
|----------|---------|----------|
| 4.0 | 10.5 | **85.5** |

We augmented the model with the $z_o$ layer described in Section 2.3 to learn a continuous speaker representation and an inference model for it. The `train-clean-{100,360}` partitions were used for training, which spans 1,172 unique speakers and, despite the name, includes many noisy recordings. As in previous experiments, by traversing each dimension of $z_l$ we found that different latent dimensions independently control different attributes of the generated speech. Moreover, this representation was disentangled from speaker identity, i.e. modifying $z_l$ did not affect the generated speaker identity if $z_o$ was fixed. In addition, we discovered that the mean of one mixture component corresponded to a narrative speaking style in a clean recording condition. Demonstrations of latent attribute control are shown in Appendix H and the demo page.

### 4.4.1 CLEAN SYNTHESIS FOR SPEAKERS WITH NOISY TRAINING DATA

We demonstrate the ability of GMVAE-Tacotron to consistently generate high-quality speech by conditioning on a value of $z_l$ associated with clean output. We considered two approaches: (1) using the mean of the identified clean component, which can be seen as a preset configuration with a fixed channel and style; (2) inferring a latent attribute representation $z_l$ from reference speech and denoising it by modifying dimensions[3] associated with the noise level to predetermined values.

We evaluated a set of eight "seen clean" (SC) speakers and a set of nine "seen noisy" (SN) speakers from the training set, a set of ten "unseen noisy" (UN) speakers from a held-out set with no overlapping speakers, and the set of ten unseen speakers used in Jia et al. (2018), denoted as "unseen clean" (UC). For consistency, we always used an inferred $z_o$ from an utterance from the target speaker, regardless of whether that speaker was seen or unseen. As a baseline we used a Tacotron model conditioned on a 128-dimensional speaker embedding learned for each speaker seen during training.

Table 3 shows the SNR of the original audio, audio synthesized by the baseline, and by the GMVAE-Tacotron using the two proposed approaches, denoted as *mean* and *latent-dn*, respectively, on all speaker sets whenever possible. In addition, to see the effectiveness of the denoising operation, the table also includes the results of using inferred $z_l$ directly, denoted as *latent*. The results show that the inferred $z_l$ followed the same SNR trend as the original audio, indicating that $z_l$ captured the variation in acoustic condition. The high SNR values of *mean* and *latent-dn* verifies the effectiveness of using a preset and denoising arbitrary inferred latent features, both of which outperformed the baseline by a large margin, and produced better quality than the original noisy audio.

Table 4 compares the proposed model using denoised $z_l$ to the baseline in a subjective side-by-side preference test. Table 5 further compares subjective naturalness MOS of the proposed model using the mean of the clean component to the baseline on the two seen speaker sets, and to the *d*-vector model (Jia et al., 2018) on the two unseen speaker sets. Specifically, we consider another stronger baseline model to compare on the SN set, which is trained on denoised data using spectral subtraction (Boll, 1979), denoted as "+ *denoise*." Both results indicate that raters preferred the proposed model to the baselines. Moreover, the MOS evaluation shows that the proposed model delivered similar level of naturalness under all conditions, seen or unseen, clean or noisy.

---

[3]We found two relevant dimensions, controlling 1) low frequency, narrowband, and 2) wideband noise levels.

Table 5: Naturalness MOS of original audio, baseline, and proposed model with the clean component mean.

| Set | Model | MOS |
|-----|-------|-----|
| SC | Original | $4.60 \pm 0.07$ |
| | Baseline | $4.17 \pm 0.07$ |
| | Proposed | $4.18 \pm 0.06$ |
| SN | Original | $4.45 \pm 0.08$ |
| | Baseline | $3.64 \pm 0.10$ |
| | + denoise | $3.84 \pm 0.10$ |
| | Proposed | $\mathbf{4.09 \pm 0.08}$ |
| UC | Original | $4.54 \pm 0.08$ |
| | $d$-vector | $4.10 \pm 0.06$ |
| | Proposed | $\mathbf{4.26 \pm 0.05}$ |
| UN | Original | $4.34 \pm 0.07$ |
| | $d$-vector | $3.76 \pm 0.12$ |
| | Proposed | $\mathbf{4.20 \pm 0.08}$ |

Table 6: Speaker similarity MOS.

| Set | Model | MOS |
|-----|-------|-----|
| SC | Baseline | $3.54 \pm 0.09$ |
| | Proposed | $3.60 \pm 0.09$ |
| SN | Original (different channels) | $3.30 \pm 0.27$ |
| | Baseline | $\mathbf{3.83 \pm 0.08}$ |
| | Baseline + denoise | $3.23 \pm 0.20$ |
| | Proposed | $3.11 \pm 0.08$ |
| UC | $d$-vector | $2.23 \pm 0.08$ |
| | $d$-vector (large) | $\mathbf{3.03 \pm 0.09}$ |
| | Proposed | $2.79 \pm 0.08$ |

### 4.4.2 SPEAKER SIMILARITY

We evaluate whether the synthesized speech resembles the identity of the reference speaker, by pairing each synthesized utterance with the reference utterance for subjective MOS evaluation of speaker similarity, following Jia et al. (2018).

Table 6 compares the proposed model using denoised latent attribute representations to baseline systems on the two seen speaker sets, and to $d$-vector systems on the unseen clean speaker set. The $d$-vector systems used a separately trained speaker encoder model to extract speaker representations for TTS conditioning as in Jia et al. (2018). We considered two speaker encoder models, one trained on the same `train-clean` partition as the proposed model, and another trained on a larger scale dataset containing 18K speakers. We denote these two systems as *d-vector* and *d-vector (large)*.

On the seen clean speaker set, the proposed model achieved similar speaker similarity scores to the baseline. However, on the seen noisy speaker set, both the proposed model and the baseline trained on denoised speech performed significantly worse than the baseline. We hypothesize that similarity of the acoustic conditions between the paired utterances biased the speaker similarity ratings. To confirm this hypothesis, we additionally evaluated speaker similarity of the ground truth utterances from a speaker whose recordings contained significant variation in acoustic conditions. As shown in Table 6, these ground truth utterances were also rated with a significantly lower MOS than the baseline, but were close to the proposed model and the denoised baseline. This result implies that this subjective speaker similarity test may not be reliable in the presence of noise and channel variation, requiring additional work to design a speaker similarity test that is unbiased to such nuisance factors.

Finally, on the unseen clean speaker set, the proposed model achieved significantly better speaker similarity scores than the $d$-vector system whose speaker representation extractor was trained on the same set as the proposed model, but worse than the $d$-vector (large) system which was trained on over 15 times more speakers. However, we emphasize that: (1) this is not a fair comparison as the two models are trained on datasets of different sizes, and (2) our proposed model is complementary to $d$-vector systems. Incorporating the high quality speaker transfer from the $d$-vector model with the strong controllability of the GMVAE is a promising direction for future work.

## 5 CONCLUSION

We describe GMVAE-Tacotron, a TTS model which learns an interpretable and disentangled latent representation to enable fine-grained control of latent attributes and provides a systematic sampling scheme for them. If speaker labels are available, we demonstrate an extension of the model that learns a continuous space that captures speaker attributes, along with an inference model which enables one-shot learning of speaker attributes from unseen reference utterances.

The proposed model was extensively evaluated on tasks spanning a wide range of signal variation. We demonstrated that it can independently control many latent attributes, and is able to cluster them without supervision. In particular, we verified using both subjective and objective tests that the model could synthesize high-quality clean speech for a target speaker even if the quality of data for that speaker does not meet high standard. These experimental results demonstrated the effectiveness of the model for training high-quality controllable TTS systems on large scale training data with rich styles by learning to factorize and independently control latent attributes underlying the speech signal.

### ACKNOWLEDGMENTS

The authors thank Daisy Stanton, Rif A. Saurous, William Chan, RJ Skerry-Ryan, Eric Battenberg, and the Google Brain, Perception and TTS teams for their helpful feedback and discussions.

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

# A    DERIVATION OF REPARAMETERIZED TRAINING OBJECTIVES

This section gives detailed derivation of the evidence lower bound (ELBO) estimation used for training. We first present a differentiable Monte Carlo estimation of the posterior $q(\mathbf{y}_l \mid \mathbf{X})$, and then derive an ELBO for each of the graphical models in Figure 1, which differ in whether an additional observed attribute representation $\mathbf{z}_o$ is used.

## A.1    MONTE CARLO ESTIMATION OF THE REPARAMETERZIED CATEGORICAL POSTERIOR

As shown in equation 2, we approximate the posterior over latent attribute class $\mathbf{y}_l$ with

$$q(\mathbf{y}_l \mid \mathbf{X}) = \mathbb{E}_{q(\mathbf{z}_l \mid \mathbf{X})}[p(\mathbf{y}_l \mid \mathbf{z}_l)], \tag{5}$$

where $q(\mathbf{z}_l \mid \mathbf{X})$ is a diagonal-covariance Gaussian, and $p(\mathbf{y}_l \mid \mathbf{z}_l)$ is the probability of $\mathbf{z}_l$ being drawn from the $\mathbf{y}_l$-th Gaussian mixture component. We first denote the mean vector and the diagonal elements of the covariance matrix of the $\mathbf{y}_l$-th component as $\boldsymbol{\mu}_{l,\mathbf{y}_l}$ and $\boldsymbol{\sigma}_{l,\mathbf{y}_l}^2$, and write the posterior over mixture components given a latent attribute representation, $p(\mathbf{y}_l \mid \mathbf{z}_l)$:

$$p(\mathbf{y}_l \mid \mathbf{z}_l) = \frac{p(\mathbf{z}_l \mid \mathbf{y}_l)p(\mathbf{y}_l)}{\sum_{\hat{\mathbf{y}}_l=1}^{K} p(\mathbf{z}_l \mid \hat{\mathbf{y}}_l)p(\hat{\mathbf{y}}_l)} \tag{6}$$

$$= \frac{f(\mathbf{z}_l; \boldsymbol{\mu}_{l,\mathbf{y}_l}, \boldsymbol{\sigma}_{l,\mathbf{y}_l}^2)K^{-1}}{\sum_{\hat{\mathbf{y}}_l=1}^{K} f(\mathbf{z}_l; \boldsymbol{\mu}_{l,\hat{\mathbf{y}}_l}, \boldsymbol{\sigma}_{l,\hat{\mathbf{y}}_l}^2)K^{-1}}, \tag{7}$$

with

$$f(\mathbf{z}_l; \boldsymbol{\mu}_{l,\mathbf{y}_l}, \boldsymbol{\sigma}_{l,\mathbf{y}_l}^2) = \frac{\exp\left\{-\frac{1}{2}(\mathbf{z}_l - \boldsymbol{\mu}_{l,\mathbf{y}_l})^\top \mathrm{diag}(\boldsymbol{\sigma}_{l,\mathbf{y}_l}^2)^{-1}(\mathbf{z}_l - \boldsymbol{\mu}_{l,\mathbf{y}_l})\right\}}{\sqrt{(2\pi)^D \left|\mathrm{diag}(\boldsymbol{\sigma}_{l,\mathbf{y}_l}^2)\right|}}, \tag{8}$$

where $D$ is the dimensionality of $\mathbf{z}_l$, and $K$ is the number of classes for $\mathbf{y}_l$.

Finally, we denote the posterior mean and variance of $q(\mathbf{z}_l \mid \mathbf{X})$ by $\hat{\boldsymbol{\mu}}_l$ and $\hat{\boldsymbol{\sigma}}_l^2$, and compute a Monte Carlo estimate of the expectation in equation 5 after reparameterization:

$$q(\mathbf{y}_l \mid \mathbf{X}) = \mathbb{E}_{q(\mathbf{z}_l \mid \mathbf{X})}[p(\mathbf{y}_l \mid \mathbf{z}_l)] \tag{9}$$

$$= \mathbb{E}_{\mathcal{N}(\boldsymbol{\epsilon}; \mathbf{0}, \boldsymbol{I})}[p(\mathbf{y}_l \mid \hat{\boldsymbol{\mu}}_l + \hat{\boldsymbol{\sigma}}_l \odot \boldsymbol{\epsilon})] \tag{10}$$

$$\approx \frac{1}{N} \sum_{n=1}^{N} p(\mathbf{y}_l \mid \hat{\boldsymbol{\mu}}_l + \hat{\boldsymbol{\sigma}}_l \odot \boldsymbol{\epsilon}^{(n)}); \quad \boldsymbol{\epsilon}^{(n)} \sim \mathcal{N}(\mathbf{0}, \boldsymbol{I}) \tag{11}$$

$$= \frac{1}{N} \sum_{n=1}^{N} \frac{f(\tilde{\mathbf{z}}_l^{(n)}; \boldsymbol{\mu}_{l,\mathbf{y}_l}, \boldsymbol{\sigma}_{l,\mathbf{y}_l}^2)K^{-1}}{\sum_{\hat{\mathbf{y}}_l=1}^{K} f(\tilde{\mathbf{z}}_l^{(n)}; \boldsymbol{\mu}_{l,\hat{\mathbf{y}}_l}, \boldsymbol{\sigma}_{l,\hat{\mathbf{y}}_l}^2)K^{-1}} \tag{12}$$

$$:= \tilde{q}(\mathbf{y}_l \mid \mathbf{X}), \tag{13}$$

where $\tilde{\mathbf{z}}_l^{(n)} = \hat{\boldsymbol{\mu}}_l + \hat{\boldsymbol{\sigma}}_l \odot \boldsymbol{\epsilon}^{(n)}$ is a random sample, drawn from a standard Gaussian distribution using $\boldsymbol{\epsilon}^{(n)} \sim \mathcal{N}(\mathbf{0}, \boldsymbol{I})$, and $N$ is the number of samples used for the Monte Carlo estimation. The resulting estimate $\tilde{q}(\mathbf{y}_l \mid \mathbf{X})$ is differentiable w.r.t. the parameters of $p(\mathbf{z}_l \mid \mathbf{y}_l)$ and $q(\mathbf{z}_l \mid \mathbf{X})$.

## A.2 DIFFERENTIABLE TRAINING OBJECTIVE

We next derive the ELBO $\mathcal{L}(p, q; \mathbf{X}, \mathbf{Y}_t, \mathbf{y}_o)$ and rewrite it as a Monte Carlo estimate used for training:

$$
\log p(\mathbf{X} \mid \mathbf{Y}_t, \mathbf{y}_o) \geq \mathbb{E}_{q(\mathbf{z}_l \mid \mathbf{X}) q(\mathbf{y}_l \mid \mathbf{X})} \left[ \log \frac{p(\mathbf{X} \mid \mathbf{Y}_t, \mathbf{y}_o, \mathbf{z}_l) p(\mathbf{z}_l \mid \mathbf{y}_l) p(\mathbf{y}_l)}{q(\mathbf{z}_l \mid \mathbf{X}) q(\mathbf{y}_l \mid \mathbf{X})} \right] \tag{14}
$$

$$
= \mathbb{E}_{q(\mathbf{z}_l \mid \mathbf{X})} \left[ \log p(\mathbf{X} \mid \mathbf{Y}_t, \mathbf{y}_o, \mathbf{z}_l) \right]
$$
$$
- \mathbb{E}_{q(\mathbf{y}_l \mid \mathbf{X})} \left[ D_{KL} \left( q(\mathbf{z}_l \mid \mathbf{X}) \,\|\, p(\mathbf{z}_l \mid \mathbf{y}_l) \right) \right]
$$
$$
- D_{KL} \left( q(\mathbf{y}_l \mid \mathbf{X}) \,\|\, p(\mathbf{y}_l) \right) \tag{15}
$$
$$
:= \mathcal{L}(p, q; \mathbf{X}, \mathbf{Y}_t, \mathbf{y}_o), \tag{16}
$$

$$
\approx \frac{1}{N'} \sum_{n'=1}^{N'} \log p(\mathbf{X} \mid \mathbf{Y}_t, \mathbf{y}_o, \tilde{\mathbf{z}}_l^{(n')})
$$
$$
- \sum_{\mathbf{y}_l=1}^{K} \tilde{q}(\mathbf{y}_l \mid \mathbf{X}) D_{KL} \left( q(\mathbf{z}_l \mid \mathbf{X}) \,\|\, p(\mathbf{z}_l \mid \mathbf{y}_l) \right)
$$
$$
- D_{KL} \left( \tilde{q}(\mathbf{y}_l \mid \mathbf{X}) \,\|\, p(\mathbf{y}_l) \right) \tag{17}
$$
$$
:= \tilde{\mathcal{L}}(p, q; \mathbf{X}, \mathbf{Y}_t, \mathbf{y}_o), \tag{18}
$$

where $\tilde{\mathbf{z}}_l^{(n')} = \hat{\boldsymbol{\mu}}_l + \hat{\boldsymbol{\sigma}}_l \odot \boldsymbol{\epsilon}^{(n')}$, $\boldsymbol{\epsilon}^{(n')} \sim \mathcal{N}(\mathbf{0}, \boldsymbol{I})$, and $\tilde{\mathcal{L}}(p, q; \mathbf{X}, \mathbf{Y}_t, \mathbf{y}_o)$ is the estimator used for training. Similarly, $N'$ is the number of samples used for the Monte Carlo estimate.

## A.3 DIFFERENTIABLE TRAINING OBJECTIVE WITH OBSERVED ATTRIBUTE REPRESENTATION

In this section, we derive the ELBO $\mathcal{L}_o(p, q; \mathbf{X}, \mathbf{Y}_t, \mathbf{y}_o)$ when using an additional observed attribute representation, $\mathbf{z}_o$, as described in Section 2.3, and rewrite it with a Monte Carlo estimation used for training. As before, we denote the posterior mean and variance of $q(\mathbf{z}_o \mid \mathbf{X})$ by $\hat{\boldsymbol{\mu}}_o$ and $\hat{\boldsymbol{\sigma}}_o^2$.

$$
\log p(\mathbf{X} \mid \mathbf{Y}_t, \mathbf{y}_o) \geq \mathbb{E}_{q(\mathbf{z}_o \mid \mathbf{X}) q(\mathbf{z}_l \mid \mathbf{X}) q(\mathbf{y}_l \mid \mathbf{X})} \left[ \log \frac{p(\mathbf{X} \mid \mathbf{Y}_t, \mathbf{z}_o, \mathbf{z}_l) p(\mathbf{z}_o \mid \mathbf{y}_o) p(\mathbf{z}_l \mid \mathbf{y}_l) p(\mathbf{y}_l)}{q(\mathbf{z}_o \mid \mathbf{X}) q(\mathbf{z}_l \mid \mathbf{X}) q(\mathbf{y}_l \mid \mathbf{X})} \right]
$$
$$
\tag{19}
$$

$$
= \mathbb{E}_{q(\mathbf{z}_o \mid \mathbf{X}) q(\mathbf{z}_l \mid \mathbf{X})} \left[ \log p(\mathbf{X} \mid \mathbf{Y}_t, \mathbf{z}_o, \mathbf{z}_l) \right]
$$
$$
- D_{KL}(q(\mathbf{z}_o \mid \mathbf{X}) \,\|\, p(\mathbf{z}_o \mid \mathbf{y}_o))
$$
$$
- \mathbb{E}_{q(\mathbf{y}_l \mid \mathbf{X})} [D_{KL}(q(\mathbf{z}_l \mid \mathbf{X}) \,\|\, p(\mathbf{z}_l \mid \mathbf{y}_l))]
$$
$$
- D_{KL}(q(\mathbf{y}_l \mid \mathbf{X}) \,\|\, p(\mathbf{y}_l)) \tag{20}
$$
$$
:= \mathcal{L}_o(p, q; \mathbf{X}, \mathbf{y}_y, \mathbf{y}_o) \tag{21}
$$

$$
\approx \frac{1}{N' N''} \sum_{n'=1}^{N'} \sum_{n''=1}^{N''} \log p(\mathbf{X} \mid \mathbf{Y}_t, \tilde{\mathbf{z}}_o^{(n')}, \tilde{\mathbf{z}}_l^{(n'')})
$$
$$
- D_{KL}(q(\mathbf{z}_o \mid \mathbf{X}) \,\|\, p(\mathbf{z}_o \mid \mathbf{y}_o))
$$
$$
- \sum_{\mathbf{y}_l=1}^{K} \tilde{q}(\mathbf{y}_l \mid \mathbf{X}) D_{KL}(q(\mathbf{z}_l \mid \mathbf{X}) \,\|\, p(\mathbf{z}_l \mid \mathbf{y}_l))
$$
$$
- D_{KL}(q(\mathbf{y}_l \mid \mathbf{X}) \,\|\, p(\mathbf{y}_l)) \tag{22}
$$
$$
:= \tilde{\mathcal{L}}_o(p, q; \mathbf{X}, \mathbf{y}_y, \mathbf{y}_o), \tag{23}
$$

where the continuous latent variables are reparameterized as $\tilde{\mathbf{z}}_o^{(n')} = \hat{\boldsymbol{\mu}}_o + \hat{\boldsymbol{\sigma}}_o \odot \boldsymbol{\epsilon}_o^{(n')}$ and $\tilde{\mathbf{z}}_l^{(n'')} = \hat{\boldsymbol{\mu}}_l + \hat{\boldsymbol{\sigma}}_l \odot \boldsymbol{\epsilon}_l^{(n'')}$, with auxiliary noise variables $\boldsymbol{\epsilon}_o^{(n')}, \boldsymbol{\epsilon}_l^{(n'')} \sim \mathcal{N}(\mathbf{0}, \boldsymbol{I})$. The estimator $\tilde{\mathcal{L}}_o(p, q; \mathbf{X}, \mathbf{Y}_t, \mathbf{y}_o)$ is used for training. $N'$ and $N''$ are the numbers of samples used for the Monte Carlo estimate.

## B    NEURAL NETWORK ARCHITECTURE DETAILS

### B.1    SYNTHESIZER

The synthesizer is an attention-based sequence-to-sequence network which generates a mel spectrogram as a function of an input text sequence and conditioning signal generated by the auxiliary encoder networks. It closely follows the network architecture of Tacotron 2 (Shen et al., 2018). The input text sequence is encoded by three convolutional layers, which contains 512 filters with shape $5 \times 1$, followed by a bidirectional long short-term memory (LSTM) of 256 units for each direction. The resulting text encodings are accessed by the decoder through a location sensitive attention mechanism (Chorowski et al., 2015), which takes attention history into account when computing a normalized weight vector for aggregation.

The base Tacotron 2 autoregressive decoder network takes as input the attention-aggregated text encoding, and the bottlenecked previous frame (processed by a pre-net comprised of two fully-connected layers of 256 units) at each step. In this work, to condition the output on additional attribute representations, the decoder is extended to consume $\mathbf{z}_l$ and $\mathbf{z}_o$ (or $\mathbf{y}_o$) by concatenating them with the original decoder input at each step. The concatenated vector forms the new decoder input, which is passed through a stack of two uni-directional LSTM layers with 1024 units. The output from the stacked LSTM is concatenated with the new decoder input (as a residual connection), and linearly projected to predict the mel spectrum of the current frame, as well as an end-of-sentence token. Finally, the predicted spectrogram frames are passed to a post-net, which predicts a residual that is added to the initial decoded sequence of spectrogram frames, to better model detail in the spectrogram and reduce the overall mean squared error.

Similar to Tacotron 2, we separately train a neural vocoder to invert a mel spectrograms to a time-domain waveform. In contrast to that work, we replace the WaveNet (van den Oord et al., 2016) vocoder with one based on the recently proposed WaveRNN (Kalchbrenner et al., 2018) architecture, which is more efficient during inference.

### B.2    LATENT ENCODER AND OBSERVED ENCODER

Both the latent encoder and the observed encoder map a mel spectrogram from a reference speech utterance to two vectors of the same dimension, representing the posterior mean and log variance of the corresponding latent variable. We design both encoders to have exactly the same architecture, whose outputs are conditioned by the decoder in a symmetric way. Disentangling of latent attributes and observed attributes is therefore achieved by optimizing different KL-divergence objectives.

For each encoder, a mel spectrogram is first passed through two convolutional layers, which contains 512 filters with shape $3 \times 1$. The output of these convolutional layers is then fed to a stack of two bidirectional LSTM layers with 256 cells at each direction. A mean pooling layer is used to summarize the LSTM outputs across time, followed by a linear projection layer to predict the posterior mean and log variance.

## C    DETAILED EXPERIMENTAL SETUP

The network is trained using the Adam optimizer (Kingma & Ba, 2015), configured with an initial learning rate $10^{-3}$, and an exponential decay that halved the learning rate every 12.5k steps, beginning after 50k steps. Parameters of the network are initialized using Xavier initialization (Glorot & Bengio, 2010). A batch size of 256 is used for all experiments. Following the common practice in the VAE literature (Kingma & Welling, 2014), we set the number of samples used for the Monte Carlo estimate to 1, since we train the model with a large batch size.

Table 7 details the list of prior hyperparameters used for each of the four datasets described in Section 4: multi-speaker English data (multi-spk), noisified multi-speaker English data (noisy-multi-spk), single-speaker story-telling data (audiobooks), and crowd-sourced audiobook data (crowd-sourced). To ensure numerical stability we set a minimum value allowed for the standard deviation of the conditional distribution $p(\mathbf{z}_l \mid \mathbf{y}_l)$. We initially set the lower bound to $e^{-1}$; however, with the exception of the multi-speaker English data, the trained standard deviation reached the lower bound

for all mixture components for all dimensions. We therefore lowered the minimum standard deviation to $e^{-2}$, and found that it left sufficient range to capture the amount of variation.

As shown in Table 11, we found that increasing the dimensionality of $\mathbf{z}_l$ from 16 to 32 improves reconstruction quality; however, it also increases the difficulty of interpreting each dimension. On the other hand, reducing the dimensionality too much can result in insufficient modeling capacity for latent attributes, however we have not carefully explored this lower bound.

Empirically, we found 16-dimensional $\mathbf{z}_l$ to be appropriate for capturing the salient attributes one would like to control in the four datasets we experimented with. When evaluating the meaning of each dimensions, we find the majority of the dimensions to be interpretable, and the number of dummy dimensions which do not affect the model output varied across datasets, as each of them inherently has variation across a different number of unlabeled attributes. For example, the model trained on the multi-speaker English corpus (Section 4.1) has four dummy dimensions of $\mathbf{z}_l$ that do not affect the output. In contrast, for the model trained on the crowd-sourced audio book corpus (Section 4.4), which contains considerably more variation in style and prosody, we found only one dummy dimension of $\mathbf{z}_l$.

Table 7: Prior hyperparameters for each dataset used in Section 4.

|  | multi-spk (Section 4.1) | noisy-multi-spk (Section 4.2) | audiobooks (Section 4.3) | crowd-sourced (Section 4.4) |
|---|---|---|---|---|
| $\dim(\mathbf{y}_l)$ | 10 | 10 | 10 | 10 |
| $\dim(\mathbf{z}_l)$ | 16 | 16 | 16 | 16 |
| initial $\boldsymbol{\sigma}_l$ | $e^0$ | $e^{-1}$ | $e^{-1}$ | $e^{-1}$ |
| minimum $\boldsymbol{\sigma}_l$ | $e^{-1}$ | $e^{-2}$ | $e^{-2}$ | $e^{-2}$ |
| $\dim(\mathbf{y}_o)$ | N/A | 84 | N/A | 1,172 |
| $\dim(\mathbf{z}_o)$ | N/A | N/A | N/A | 16 |
| initial $\boldsymbol{\sigma}_o$ | N/A | N/A | N/A | $e^{-2}$ |
| minimum $\boldsymbol{\sigma}_o$ | N/A | N/A | N/A | $e^{-4}$ |

## D  POSTERIOR COLLAPSE

There are two latent variables in our graphical model as shown in Figure 1(left): the latent attribute class $\mathbf{y}_l$ (discrete) and latent attribute representation $\mathbf{z}_l$ (continuous). We discuss the potential for posterior collapse for each of them separately.

### D.1  POSTERIOR COLLAPSE OF LATENT ATTRIBUTE REPRESENTATION

The continuous latent variable $\mathbf{z}_l$ is used to directly condition the generation of $\mathbf{X}$, along with two other observed variables, $\mathbf{y}_o$ and $\mathbf{Y}_t$. In our experiments, we observed that the latent variable $\mathbf{z}_l$ is always used, i.e. the KL-divergence of $\mathbf{z}_l$ never drops to zero, without applying any tricks such as KL-annealing (Bowman et al., 2016).

Previous studies report posterior-collapse of directly conditioned latent variables when using strong models (e.g. auto-regressive networks) to parameterize the conditional distribution of text (Bowman et al., 2016; Yang et al., 2017; Kim et al., 2018). This phenomenon arises from the competition between (1) increasing reconstruction performance by utilizing information provided by the latent variable, and (2) decreasing the KL-divergence by making the latent variable uninformative. Auto-regressive models are more likely to converge to the second case during training because the improvement in reconstruction from utilizing the latent variable can be smaller than the increase in KL-divergence.

However, this does not always happen, because the amount of improvement resulted from utilizing the information provided by the latent variable depends on the type of data. The reason that the posterior-collapse does not occur in our experiments is likely a consequence of the complexity of the speech sequence distribution, compared to text. Even though we use an auto-regressive decoder,

reconstruction performance can still be improved significantly by utilizing the information from $z_l$, and such improvement overpowers the increase in KL-divergence.

## D.2 POSTERIOR COLLAPSE OF LATENT ATTRIBUTE CLASS

The discrete latent variable $\mathbf{y}_l$ indexes the mixture components in the space of latent attribute representation $\mathbf{z}_l$. We did not observe the phenomenon of degenerate clusters mentioned in Dilokthanakul et al. (2016) when training our model using the hyperparameters listed in Table 7. Below we identify the difference between our GMVAE and that in Dilokthanakul et al. (2016), which we will refer to as Dil-GMVAE, and explain why our formulation is less likely to suffer from similar posterior collapse issues.

In our model, the conditional distribution $p(\mathbf{z}_l \mid \mathbf{y}_l) = \mathcal{N}\left(\boldsymbol{\mu}_{\mathbf{y}_l}, \text{diag}(\boldsymbol{\sigma}_{\mathbf{y}_l})\right)$ is a diagonal-covariance Gaussian, parameterized by a mean and a covariance vector. In contrast, in Dil-GMVAE, the conditional distribution of $\mathbf{z}_l$ given $\mathbf{y}_l$ is much more flexible, because it is parameterized using neural networks as:

$$p(\mathbf{z}_l \mid \mathbf{y}_l) = \int_{\boldsymbol{\epsilon}} \mathcal{N}(\mathbf{z}_l; f_{\boldsymbol{\mu}}(\mathbf{y}_l, \boldsymbol{\epsilon}), f_{\boldsymbol{\sigma}^2}(\mathbf{y}_l, \boldsymbol{\epsilon})) \, \mathcal{N}(\boldsymbol{\epsilon}; \mathbf{0}, \boldsymbol{I}) \, d\boldsymbol{\epsilon}, \tag{24}$$

where $f_{\boldsymbol{\mu}}$ and $f_{\boldsymbol{\sigma}^2}$ are neural networks that take $\mathbf{y}_l$ and an auxiliary noise variable $\boldsymbol{\epsilon}$ as input to predict the mean and variance of $p(\mathbf{z}_l \mid \mathbf{y}_l, \boldsymbol{\epsilon})$, respectively. The conditional distribution of each component in Dil-GMVAE can be seen as a mixture of infinitely many diagonal-covariance Gaussian distributions, which can model much more complex distributions, as shown in Dilokthanakul et al. (2016, Figure 2(d)). Compared with the GMVAE described in this paper, Dil-GMVAE can be regarded as having a much stronger stochastic decoder that maps $\mathbf{y}_l$ to $\mathbf{z}_l$.

Suppose the conditional distribution that maps $\mathbf{z}_l$ to $\mathbf{X}$ can benefit from having a very complex, non-Gaussian marginal distribution over $\mathbf{z}_l$, denoted as $p^*(\mathbf{z}_l)$. To obtain such a marginal distribution of $\mathbf{z}_l$ in Dil-GMVAE, the stochastic decoder that maps $\mathbf{y}_l$ to $\mathbf{z}_l$ can choose between (1) using the same $p(\mathbf{z}_l \mid \mathbf{y}_l) = p^*(\mathbf{z}_l)$ for all $\mathbf{y}_l$, or (2) having $p(\mathbf{z}_l \mid \mathbf{y}_l)$ model a different distribution for each $\mathbf{y}_l$, and $\sum_{\mathbf{y}_l} p(\mathbf{z}_l \mid \mathbf{y}_l) \, p(\mathbf{y}_l) = p^*(\mathbf{z}_l)$. As noted in Dilokthanakul et al. (2016), the KL-divergence term for $\mathbf{y}_l$ in the ELBO prefers the former case of degenerate clusters that all model the same distribution. As a result, the first option would be preferred with respect to the ELBO objective compared to the second one, because it does not compromise the expressiveness of $\mathbf{z}_l$ while minimizing the KL-divergence on $\mathbf{y}_l$. In contrast, our GMVAE formulation reduces to a single Gaussian when $p(\mathbf{z}_l \mid \mathbf{y}_l)$ is the same for all $\mathbf{y}_l$, and hence there is a trade-off between the expressiveness of $p(\mathbf{z}_l)$ and the KL-divergence on $\mathbf{y}_l$.

In addition, we now explain the connection between posterior-collapse and hyperparameters of the conditional distribution $p(\mathbf{z}_l \mid \mathbf{y}_l)$ in our work. In our GMVAE model, posterior-collapse of $\mathbf{y}_l$ is equivalent to having the same conditional mean and variance for each mixture component. In the ELBO derived from our model, there are two terms that are relevant to $p(\mathbf{z}_l|\mathbf{y}_l)$, which are (1) the expected KL-divergence on $\mathbf{z}_l$: $\mathbb{E}_{q(\mathbf{y}_l|\mathbf{X})}\left[D_{KL}(q(\mathbf{z}_l|\mathbf{X})||p(\mathbf{z}_l|\mathbf{y}_l))\right]$ and (2) the KL-divergence on $\mathbf{y}_l$: $D_{KL}(q(\mathbf{y}_l|\mathbf{X})||p(\mathbf{y}_l))$. The second term encourages a uniform posterior $q(\mathbf{y}_l|\mathbf{X})$, which effectively pulls the conditional distribution for each component to be close to each other, and promotes posterior collapse. In contrast, the first term pulls each $p(\mathbf{z}_l|\mathbf{y}_l)$ to be close to $q(\mathbf{z}_l|\mathbf{X})$ with a force proportional to the posterior of that component, $q(\mathbf{y}_l|\mathbf{X})$.

In one extreme, where the posterior $q(\mathbf{y}_l|\mathbf{X})$ is close to uniform, each $p(\mathbf{z}_l|\mathbf{y}_l)$ is also pushed toward the same distribution, $q(\mathbf{z}_l|\mathbf{X})$, which promotes posterior collapse. In the other extreme, where the posterior $q(\mathbf{y}_l|\mathbf{X})$ is close to one-hot with $q(\mathbf{y}_l = k|\mathbf{X}) \approx 1$, only the conditional distribution of the assigned component $p(\mathbf{z}_l|\mathbf{y}_l = k)$ is pushed toward $q(\mathbf{z}_l|\mathbf{X})$. As long as different $\mathbf{X}$ are assigned to different components, this term is anti-collapse. Therefore, we can see that the effect of the first term on posterior-collapse depends on the entropy of $q(\mathbf{y}_l|\mathbf{X})$, which is controlled by the scale of the variance when the means are not collapsed. This variance is similar to the temperature parameter used in softmax: the smaller the variance is, the more spiky the posterior distribution over y is. This is why we set the initial variance of each component to a smaller value at the beginning of training, which helps avoid posterior collapse

# E  ADDITIONAL RESULTS ON THE MULTI-SPEAKER ENGLISH CORPUS

## E.1  RANDOM SAMPLES BY MIXTURE COMPONENT

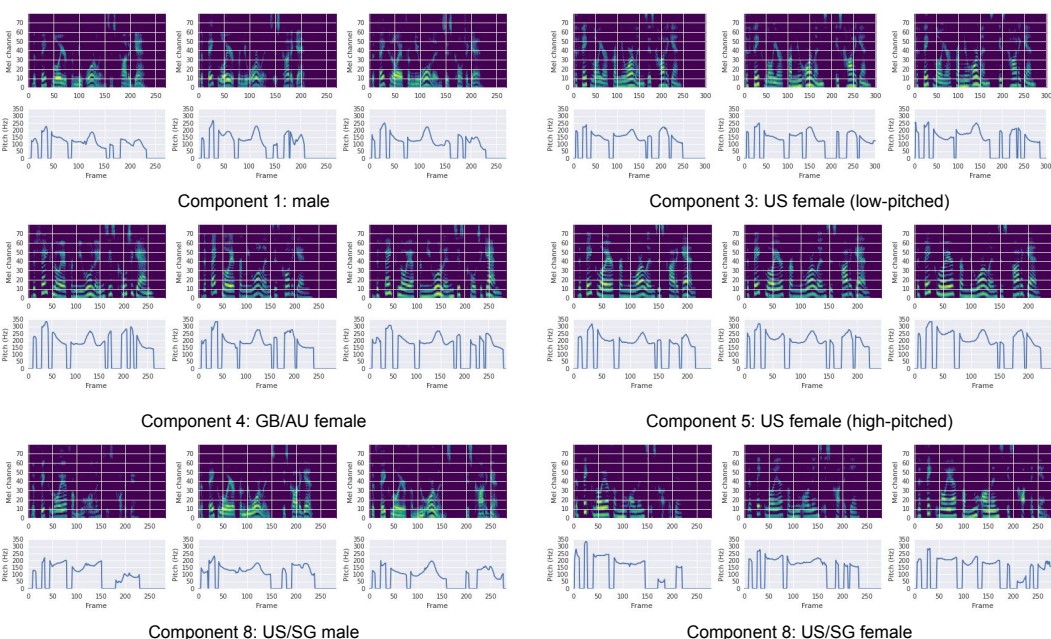

Figure 8: Mel-spectrograms and $F_0$ tracks of three random samples drawn from each of six selected mixture components. Each component represents certain gender and accent group. The input text is "The fake lawyer from New Orleans is caught again." which emphasizes the difference between British and US accents. As mentioned in the paper, although samples from component 3 and 5 both capture US female voices, each component captures specific speakers with different $F_0$ ranges. The former ranges from 100 to 250 Hz, and the latter ranges from 200 to 350 Hz. Audio samples can be found at https://google.github.io/tacotron/publications/gmvae_controllable_tts#multispk_en.sample

## E.2  QUANTITATIVE ANALYSIS OF THE US FEMALE COMPONENTS

To quantify the difference between the three components that model US female speakers (3, 5, and 6), we draw 20 latent attribute encodings $z_l$ from each of the three components and decode the same set of 25 text sequences for each one. Table 8 shows the average $F_0$ computed over 500 synthesized utterances for each component, demonstrating that each component models a different $F_0$ range.

Table 8: $F_0$ distribution for the three US female components.

| Component | 3 | 5 | 6 |
|---|---|---|---|
| $F_0$ (Hz) | $169.1 \pm 10.1$ | $214.8 \pm 8.4$ | $192.2 \pm 21.4$ |

### E.3   CONTROL OF LATENT ATTRIBUTES

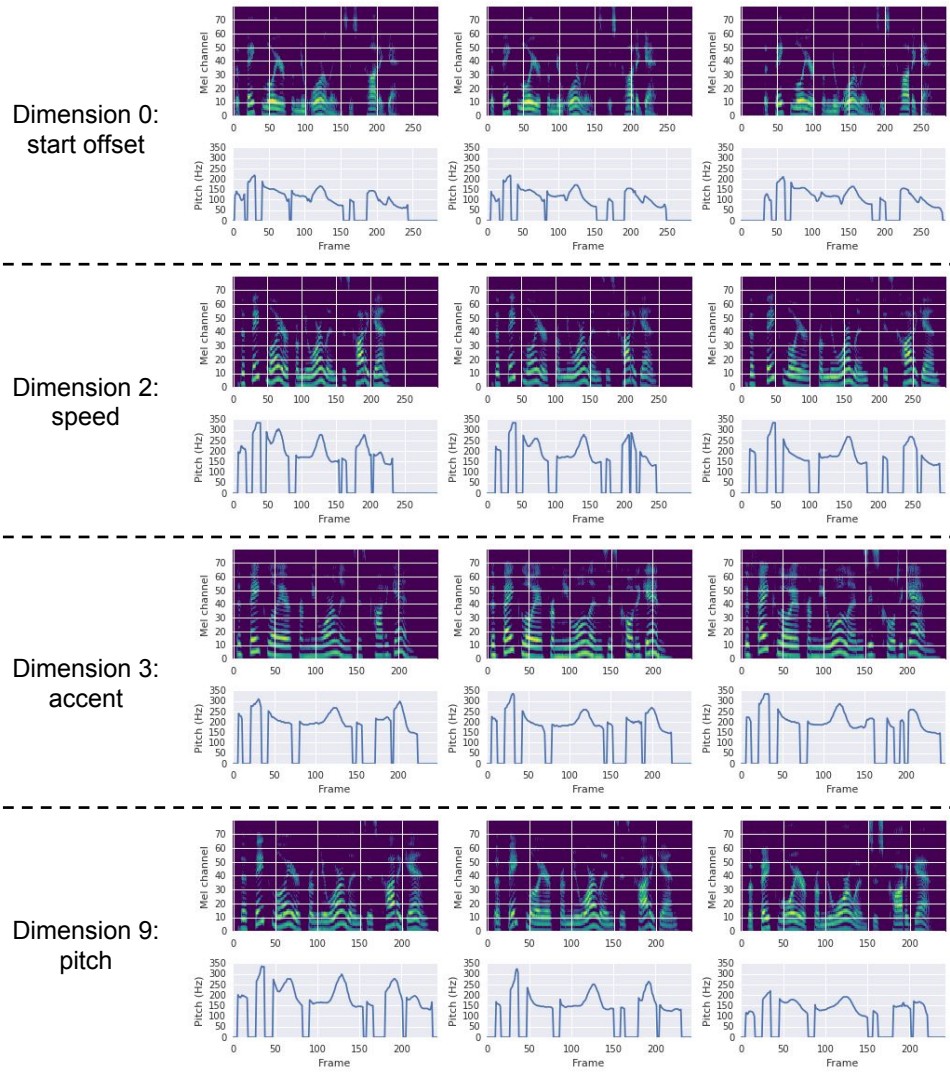

Figure 9: Mel-spectrograms and $F_0$ tracks of the synthesized samples demonstratoing independent control of several latent attributes. Each row traverses one dimension with three different values, keeping all other dimensions fixed.. All examples use the same input text: "The fake lawyer from New Orleans is caught again." The plots for dimension 0 (top row) and dimension 2 (second row) mainly show variation along the time axis. The underlying $F_0$ contour values do not change, however dimension 0 controls the duration of the initial pause before the speech begins, and dimension 2 controls the overall speaking rate, with the $F_0$ track stretching in time (i.e. slowing down) when moving from the left column to the right. Dimension nine (bottom row) mainly controls the degree of $F_0$ variation while maintaining the speed and starting offset. Finally, we note that differences in accent controlled by dimension 3 (third row) are easier to recognize by listening to audio samples, which can be found at `https://google.github.io/tacotron/publications/gmvae_controllable_tts#multispk_en.control`.

### E.4 CLASSIFICATION OF LATENT ATTRIBUTE REPRESENTATIONS

Table 9: Accuracy (%) of linear classifiers trained on $\mathbf{z}_l$.

|       | Gender | Accent | Speaker Identity |
|-------|--------|--------|------------------|
| Train | 100.00 | 98.76  | 97.66            |
| Eval  | 98.72  | 98.72  | 95.39            |

To quantify how well the learned representation captures useful speaker information, we experimented with training classifiers for speaker attributes on the latent features. The test utterances were partitioned in a 9:1 ratio for training and evaluation, which contain 2,098 and 234 utterances, respectively. Three linear discriminant analysis (LDA) classifiers were trained on the latent attribute representations $\mathbf{z}_l$ to predict speaker identity, gender and accent. Table 9 shows the classification results. The high accuracies in the table demonstrate the potential of applying the learned low-dimensional representations to tasks of predicting other unseen attributes.

## F ADDITIONAL RESULTS ON THE NOISY MULTI-SPEAKER ENGLISH CORPUS

### F.1 RANDOM SAMPLES FROM NOISY AND CLEAN COMPONENTS

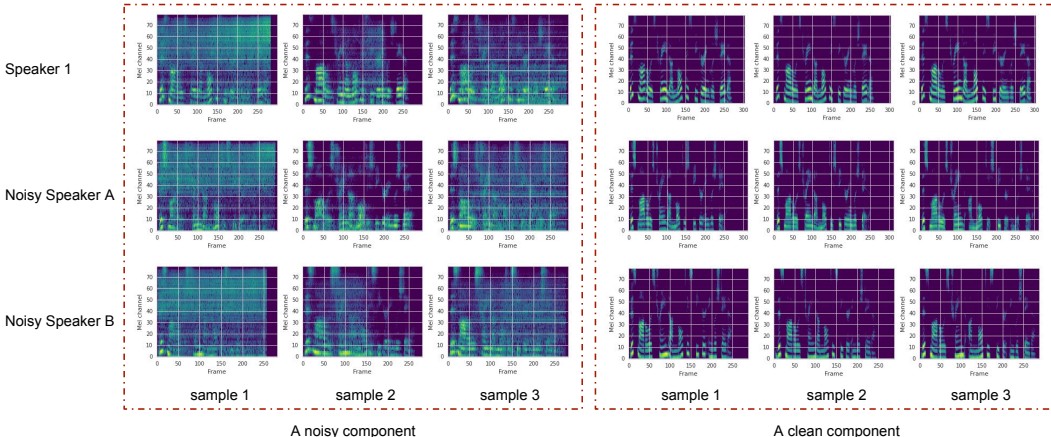

Figure 10: Mel-spectrograms of random samples drawn from a noisy (left) and a clean (right) mixture component. Samples within each row are conditioned on the same speaker. Likewise, samples within each column are conditioned on the same latent attribute representation $\mathbf{z}_l$. For all samples, the input text is "This model is trained on multi-speaker English data." Samples drawn from the clean component are all clean, while samples drawn from the noisy component all contain obvious background noise. Finally, note that samples within each column contain similar types of noise since they are conditioned on the same $\mathbf{z}_l$. Audio samples can be found at `https://google.github.io/tacotron/publications/gmvae_controllable_tts#noisy_multispk_en.sample`

## F.2 CONTROL OF BACKGROUND NOISE LEVEL

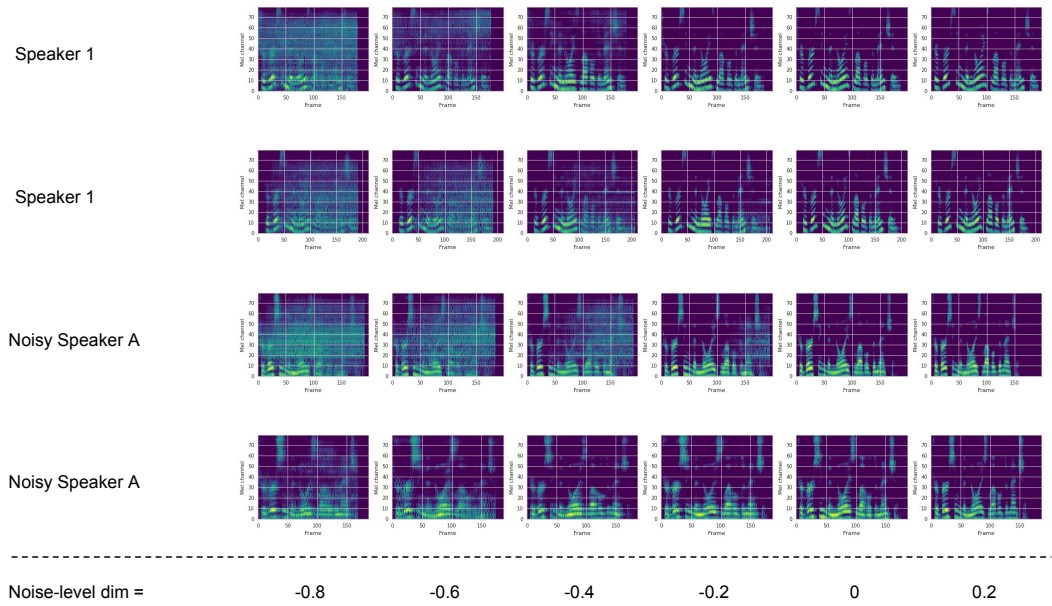

Figure 11: Mel-spectrograms of the synthesized samples demonstrating control of the background noise level by varying the value of dimension 13. Each row conditions on a seed $z_l$ drawn from a mixture component, where all values except for dimension 13 are fixed. The embedding used in row 1 and row 3 are drawn from a noisy component, and used in row 2 and row 4 are drawn from a clean component. In addition, we condition the decoding on the same speaker for the first two rows, and the same held-out speaker for the last two rows. The value of dimension 13 used in each column is shown at the bottom, and the input text is "Traversing the noise level dimension." In all rows, samples on the right are cleaner than those on the left, with the background noise gradually fading away as the value for dimension 13 increases. Audio samples can be found at `https://google.github.io/tacotron/publications/gmvae_controllable_tts#noisy_multispk_en.control`

## F.3 PRIOR DISTRIBUTION OF THE NOISE LEVEL DIMENSION

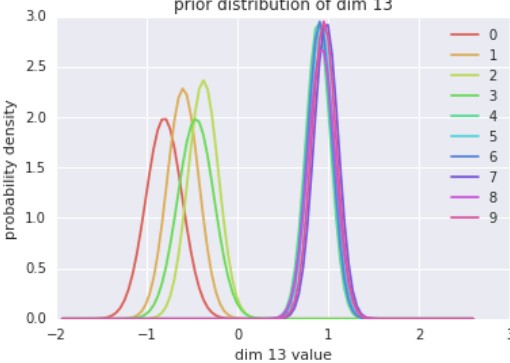

Figure 12: Prior distributions of each component for dimension 13, which controls background noise level. The first four components (0–3) model noisy speech, and the other six (4–9) model clean speech. The two groups of mixture components are clearly separated in this dimension. Furthermore, the clean components have lower variances than the noisy components, indicated a narrower range of noise levels in clean components compared to noisy ones.

## G ADDITIONAL RESULTS ON THE SINGLE-SPEAKER AUDIOBOOK CORPUS

### G.1 QUANTITATIVE EVALUATION ON DISENTANGLED CONTROL

To objectively evaluate the ability of the proposed GMVAE-Tacotron model to control individual attributes, we compute two metrics: (1) the average $F_0$ (fundamental frequency) in voiced frames, computed using YIN (De Cheveigné & Kawahara, 2002), and (2) the average speech duration. These correspond to rough measures of the speaking rate and degree of pitch variation, respectively.

We randomly draw 10 samples of seed $\mathbf{z}_l$ from the prior, deterministically set the target dimension to $\bar{\mu}_{l,d} - 3\bar{\sigma}_{l,d}$, $\bar{\mu}_{l,d}$, and $\bar{\mu}_{l,d} + 3\bar{\sigma}_{l,d}$ to construct modified $\mathbf{z}_l^*$, where $\bar{\mu}_{l,d}$ and $\bar{\sigma}_{l,d}$ are the mean and standard deviation of the marginal distribution of the target dimension $d$. We then synthesize a set of the same 25 text sequences for each of the 30 resulting values of $\mathbf{z}_l^*$. For each value of the target dimension, we compute an average metric over 250 synthesized utterances (10 seed $\mathbf{z}_l \times 25$ text inputs).

Table 10: Quantitative evaluation of disentangled attribute control of GMVAE-Tacotron.

| Target dim | Attribute | $\bar{\mu}_{l,d} - 3\bar{\sigma}_{l,d}$ | $\bar{\mu}_{l,d}$ | $\bar{\mu}_{l,d} + 3\bar{\sigma}_{l,d}$ |
|---|---|---|---|---|
| Pitch dimension ($d = 8$) | Pitch (Hz) | 178.3 | 187.1 | 204.4 |
| | Duration (seconds) | 3.5 | 3.5 | 3.5 |
| Speaking rate dimension ($d = 7$) | Pitch (Hz) | 197.1 | 189.7 | 182.5 |
| | Duration (seconds) | 3.0 | 3.5 | 4.0 |

Results are shown in Table 10. For the "pitch" dimension, we can see that the measured $F_0$ varies substantially, while the speech remains constant. Similarly, as the value in the "speaking rate" dimension varies, the measured duration varies substantially. However, there is also a smaller inverse effect on the pitch, i.e. the pitch slightly increases with the speaking rate, which is consistent with natural speaking behavior (Apple et al., 1979; Black, 1961). These results are an indication that manipulating individual dimensions primarily controls the corresponding attribute.

### G.2 PARALLEL STYLE TRANSFER

We evaluated the ability of the proposed model to synthesize speech that resembled the prosody or style of a given reference utterance, by conditioning on a latent attribute representation inferred from the reference. We adopted two metrics from Skerry-Ryan et al. (2018) to quantify style transfer performance: the mel-cepstral distortion(MCD$_{13}$), measuring the phonetic and timbral distortion, and $F_0$ frame error (FFE), which combines voicing decision error and $F_0$ error metrics to capture how well $F_0$ information, which encompasses much of the prosodic content, is retained. Both metrics assume that the generated speech and the reference speech are frame aligned. We therefore synthesized the same text content as the reference for this evaluation.

Table 11: Quantitative evaluation for parallel style transfer. Lower is better for both metrics.

| Model | MCD$_{13}$ | FFE |
|---|---|---|
| Baseline | 17.91 | 64.1% |
| GST | **14.34** | **41.0%** |
| Proposed (16) | 15.78 | 51.4% |
| Proposed (32) | 14.42 | 42.5% |

Table 11 compares the proposed model against the baseline and a 16-token GST model. The proposed model with a 16-dimensional $\mathbf{z}_l$ ($D = 16$) was better than the baseline but inferior to the GST model. Because the GST model uses a four-head attention (Vaswani et al., 2017), it effectively has 60 degrees of freedom, which might explain why it performs better in replicating the reference style. By increasing the dimension of $\mathbf{z}_l$ to 32 ($D = 32$), the gap to the GST model is greatly reduced. Note

that the number of degrees of freedom of the latent space as well as the total number of parameters is still smaller than in the GST model. As noted in Skerry-Ryan et al. (2018), the ability of a model to reconstruct the reference speech frame-by-frame is highly correlated with the latent space capacity, and we can expect our proposed model to outperform GST on these two metrics if we further increase the dimensionality of $\mathbf{z}_l$; however, this would likely reduce interpretability and generalization of the latent attribute control.

### G.3 Non-parallel style transfer

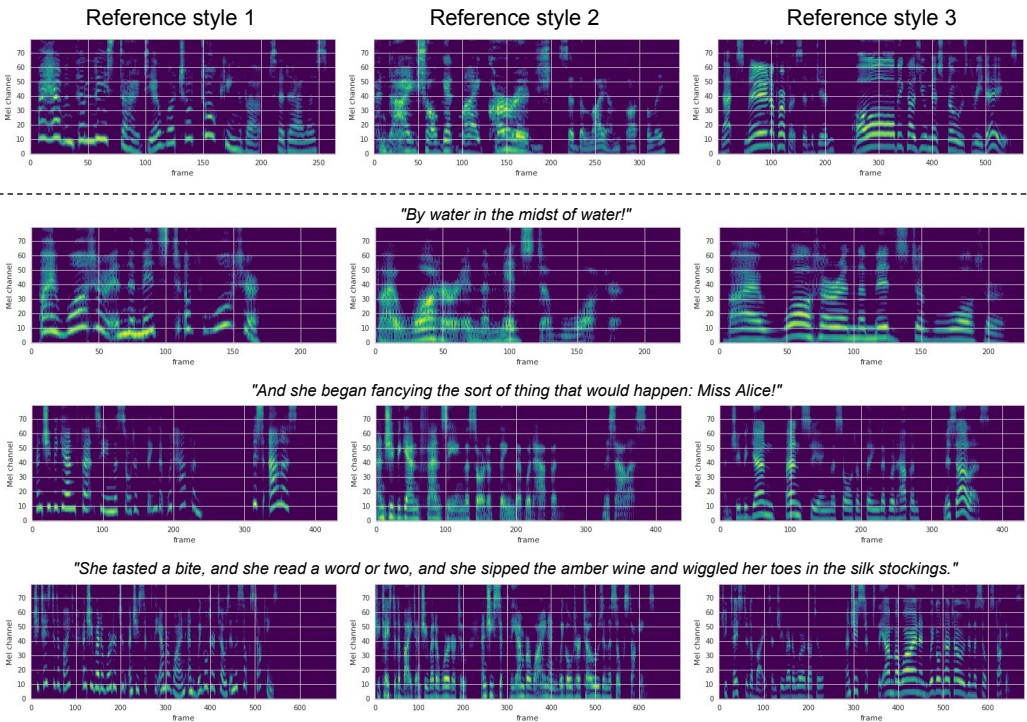

Figure 13: Mel-spectrograms of reference and synthesized style transfer utterances. The four reference utterances are shown on the top, and the four synthesized style transfer samples are shown below, where each row uses the same input text (shown above the spectrograms), and each column is conditioned on the $\mathbf{z}_l$ inferred from the reference in the top row. From left to right, the voices of the three reference utterances can be described as (1) tremulous and high-pitched, (2) rough, low-pitched, and terrifying, and (3) deep and masculine. In all cases, the synthesized samples resemble the prosody and the speaking style of the reference. For example, samples in the first column have the highest $F_0$ (positively correlated to the spacing between horizontal stripes) and more tremulous (vertical fluctuations), and spectrograms in the middle column are more blurred, related to roughness of a voice. Audio samples can be found at `https://google.github.io/tacotron/publications/gmvae_controllable_tts#singlespk_audiobook.transfer`

Figure 13 demonstrates that the GMVAE-Tacotron can also be applied in a non-parallel style transfer scenario to generate speech whose text content differs significantly from the reference.

## G.4 RANDOM STYLE SAMPLES

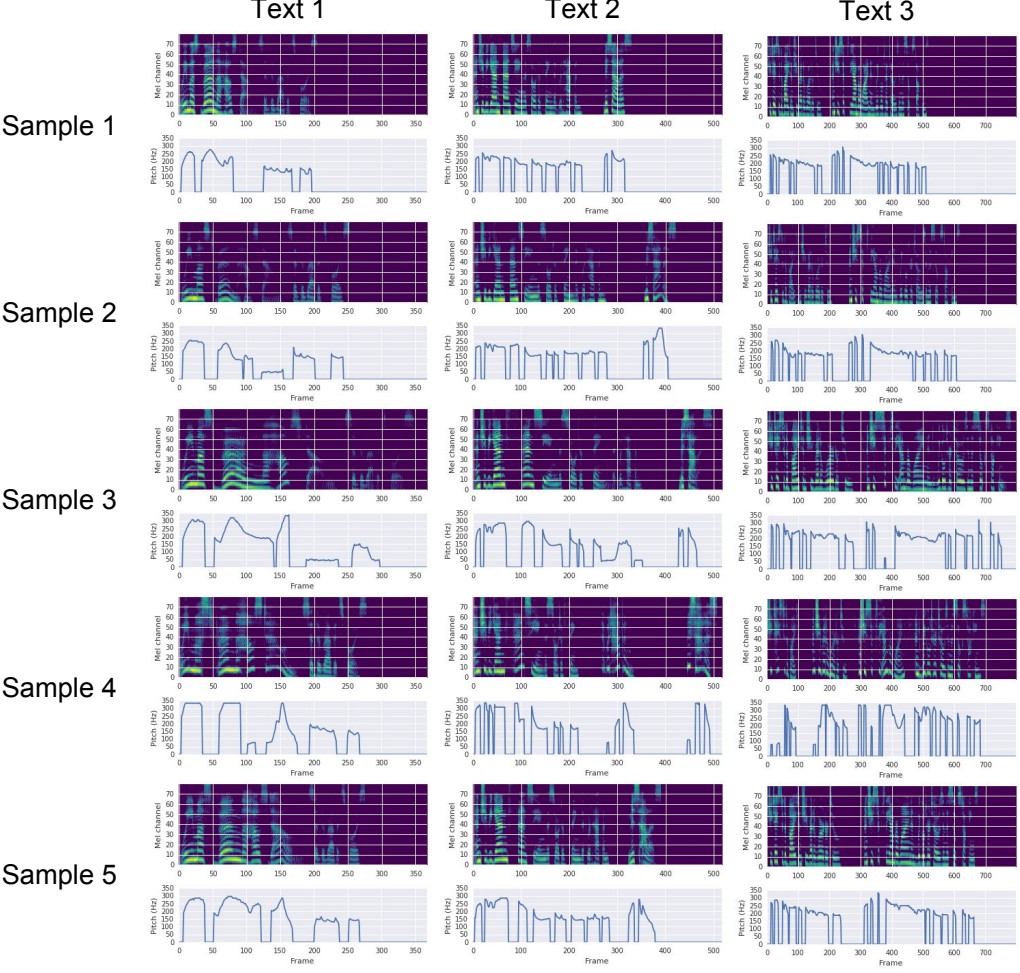

Figure 14: Mel-spectrograms and $F_0$ tracks of different input text with five random samples of $\mathbf{z}_l$ drawn from the prior. The three input text sequences from left to right are: (1) "We must burn the house down! said the Rabbit's voice.", (2) "And she began fancying the sort of thing that would happen: Miss Alice!", and (3) "She tasted a bite, and she read a word or two, and she sipped the amber wine and wiggled her toes in the silk stockings." The five samples of $\mathbf{z}_l$ encode different styles: the first sample has the fastest speaking rate, the third sample has the slowest speaking rate, and the fourth sample has the highest $F_0$. Audio samples can be found at `https://google.github.io/tacotron/publications/gmvae_controllable_tts#singlespk_audiobook.sample`

## G.5 CONTROL OF STYLE ATTRIBUTES

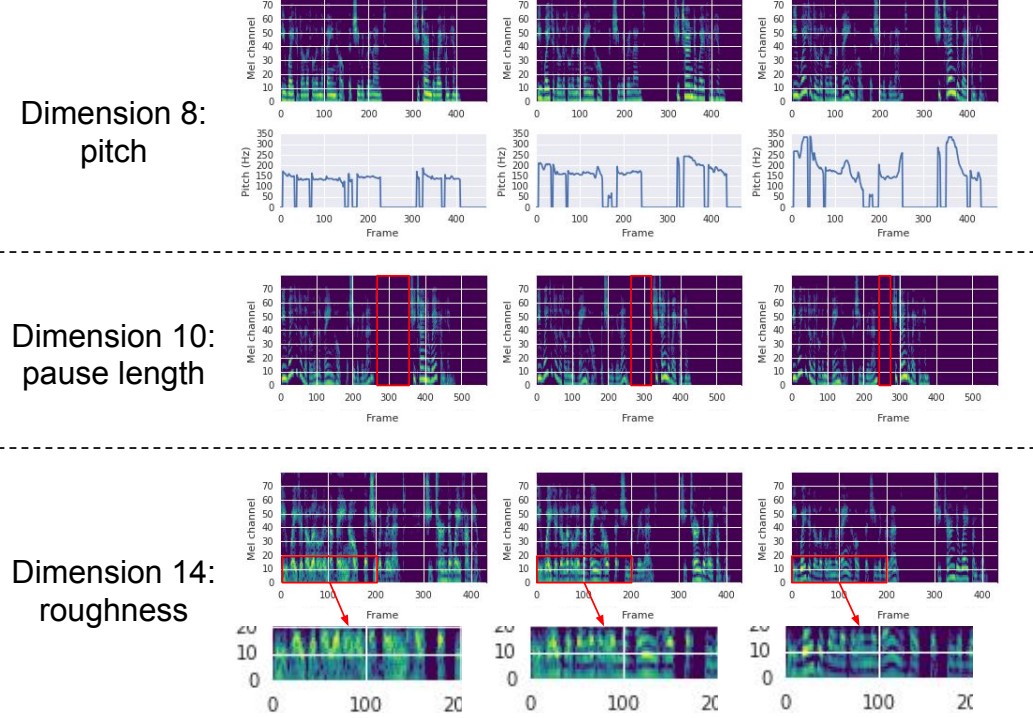

Figure 15: Synthesized mel-spectrograms demonstrating independent control of speaking style and prosody. The same input text is used for all samples: "He waited a little, in the vain hope that she would relent: she turned away from him." In the top row, $F_0$ is controlled by setting different values for dimension eight. $F_0$ tracks show that the $F_0$ range increases from left to right, while other attributes such as speed and rhythm do not change. In the second row, the duration of pause before the phrase "she turned away from him." (red boxes) is varied. The three spectrograms are very similar, except for the width of the red boxes, indicating that only the pause duration changed. In the bottom row, the "roughness" of the voice is varied. The same region of spectrograms is zoomed-in for clarity, where the spectrograms became less blurry and the harmonics becomes better defined from left to right. Audio samples can be found at https://google.github.io/tacotron/publications/gmvae_controllable_tts#singlespk_audiobook.control.

# H ADDITIONAL RESULTS ON THE CROWD-SOURCED AUDIOBOOK CORPUS

## H.1 CONTROL OF STYLE, CHANNEL, AND NOISE ATTRIBUTES

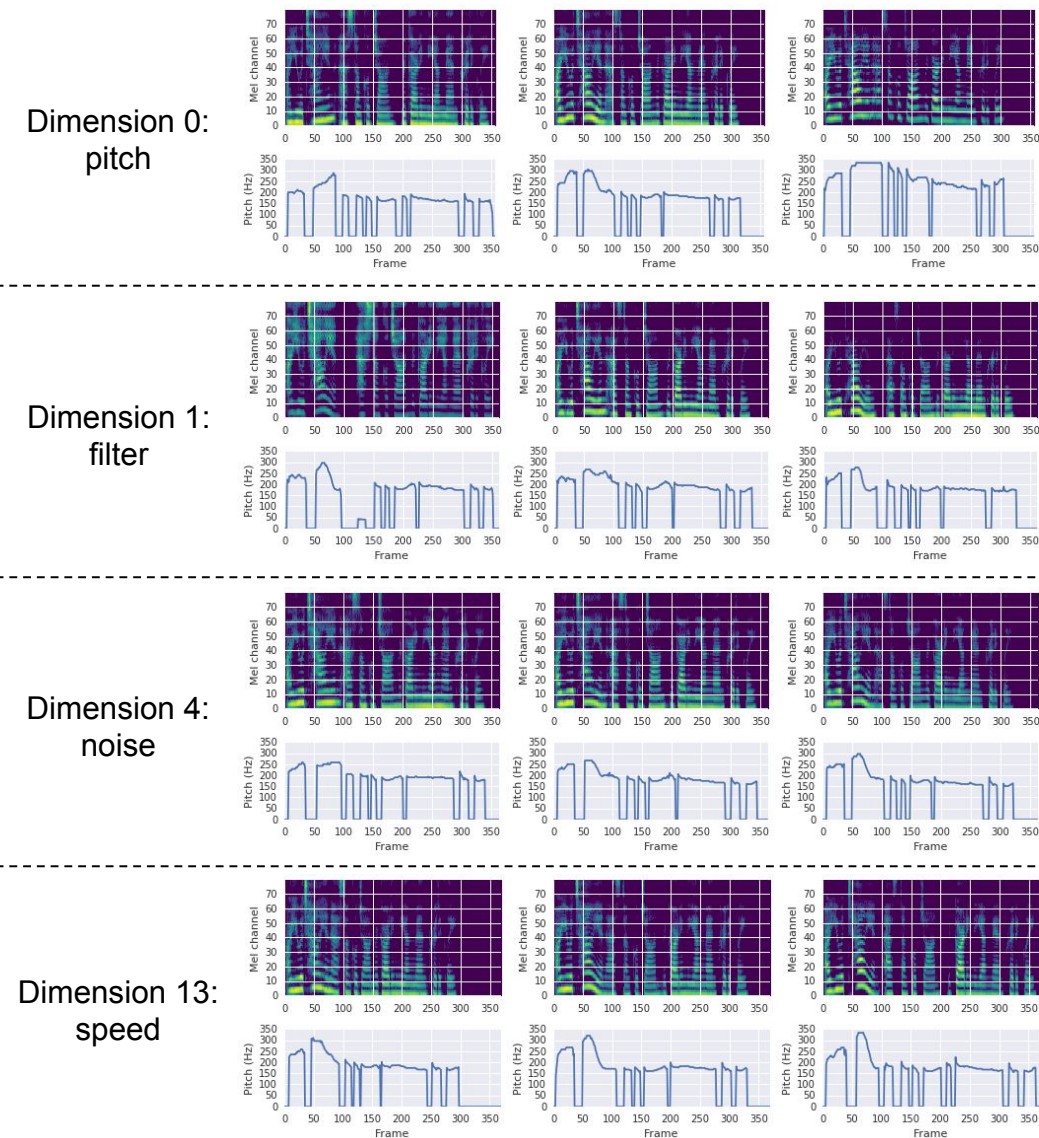

Figure 16: Synthesized mel-spectrograms and $F_0$ tracks demonstrating independent control of attributes related to style, recording channel, and noise-condition. The same text input was used for all the samples: '"Are you Italian?" asked Uncle John, regarding the young man critically.' In each row we varied the value for a single dimension while holding other dimensions fixed. In the top row, we controlled the $F_0$ by traversing dimension zero. Note that the speaker identity did not change while traversing this dimension. In the second row, the $F_0$ contours did change while traversing this dimension; however, it can be seen from the spectrograms that the leftmost one attenuated the energy in low-frequency bands, and the rightmost one attenuated energy in high-frequency bands. This dimension appears to control the shape of a linear filter applied to the signal, perhaps corresponding to variation in microphone frequency response in the training data. In the third row, the $F_0$ contours did not change, either. However, the background noise level does vary while traversing this dimension, which can be heard on the demo page. In the bottom row, variation in the speaking rate can be seen while other attributes remain constant. Audio samples can be found at `https://google.github.io/tacotron/publications/gmvae_controllable_tts#crowdsourced_audiobook.control`.

