# OpenReview forum: "Hierarchical Generative Modeling for Controllable Speech Synthesis"
_ICLR.cc/2019/Conference_

### Official Review · AnonReviewer1 · 2018-11-03
**Hierarchical Generative Modeling for Controllable Speech Synthesis**

**Rating:** 5
**Confidence:** 4

**Review:**

Quality: This submission claims to present a model that can control non-annotated attributes such as speaking style, accent, background noise, etc. Though empirical evidence in the form of numerical measurements is presented for some controllable attributes more evidence other than individual samples and authors claims is needed. For example a reliable numerical evidence is needed on page 4 following "We also found...", page 5 following "We discovered....", page 5 following "It clearly presents...", page 5 following "Drawing samples..." evidence is given only for 1 dimension, page 6 following "Figure 7(b)...".

Clarity: The model is simple though the exact form and nature of observed and latent class variables could be made more explicit. Including how they are computed/initialised/set. What are different modes using the proposed model? Why both negative results are in the appendix?

Originality: moderately

Significance: moderately

---

> ### Author Response · Authors · 2018-11-14
> **Response to AnonReviewer1 (Part 1)**
>
> We thank the reviewer for the great feedback. Below are the itemized responses regarding each comment. We will also incorporate these into the revised version.
>
> Due to the "max 5000 characters" limit, we break down our response into multiple comments.
>
>
> Re:
> Though empirical evidence in the form of numerical measurements is presented for some controllable attributes more evidence other than individual samples and authors claims is needed.
>
> Ans:
> - Identifying reliable objective quality metrics for TTS, and indeed generative models more generally, is an open problem. This is especially true for high level style concepts such as accent, or labels related to emotional valence or arousal. It is common practice across papers on generating real data to demonstrate the ability to independently control attributes qualitatively, using generated samples, providing quantitative evaluation on few attributes that are easy to estimate due to similar difficulties, e.g. Glow (Kingma et al., 2018). Qualitative evaluation similar to what we provide has been the standard in other recent work on controllable TTS models such as GST (Wang et al., 2018) and VAE-Loop (Akuzawa et al., 2018).
> - In addition, it is difficult to provide comprehensive quantitative evaluation because (1) the set of unlabeled attributes we are able to measure is limited to those for which reliable systems exist to estimate them from the speech signal, as in e.g. pitch tracking. It is unclear how to convincingly measure the level of excitement, deepness, or roughness of a voice for style control. (2) Also, estimation of attributes such as pitch can be prone to error. For example, pitch estimation of rough voices is not accurate because the harmonic patterns are not as clear as normal voices (See the bottom row in Figure 16). Even in cases where it is straightforward to report statistics to show the attribute we claim to be affected does indeed vary, simple metrics alone do not show the whole picture since they do not verify that other attributes remain the same. However, such *independent* control of an individual attribute is easily demonstrated qualitatively. To circumvent the difficulty, we provide multiple samples on the demo page to demonstrate that control over each attribute is invariant to other decoder input.
>
>
> Re:
> page 5 following "It clearly presents the samples (in fact, all the samples) drawn from components in group one were noisy, while the samples drawn from the other components were clean"
>
> Ans:
> - Quantitative evaluation on the ability to control noise level is presented in Figure 5, Figure 13, and Table 1.
> -- In Figure 5, we plot the WADA-SNR with respect to different values for the noise level dimension, which demonstrates the ability to control noise level through that dimension. In Figure 13, we plot the distribution of the noise level dimension for each mixture component, showing two groups of components that are well separated in this dimension. We can conclude that the group of mixture components on the right in Figure 13 generate clean speech because they have high probability of generating samples that correspond to high SNRs. In contrast, the group on the left in Figure 13 generates noisy speech.
> -- In Table 1, we report WADA-SNR when conditioning on the mean of a clean component, verifying that the clean component can consistently generate clean speech.

---

> > ### Author Response · Authors · 2018-11-14
> > **Response to AnonReviewer1 (Part 2)**
> >
> >
> > Re:
> > The model is simple though the exact form and nature of observed and latent class variables could be made more explicit. Including how they are computed/initialised/set.
> > Ans:
> > - Throughout the paper, we use y_* to denote discrete class variables, and z_* to denote continuous variables. We attempted to explain y_l clearly In Section 2.1 of the text and in Figure 1, e.g. it says “y_l is a K-way categorical discrete variable” that identifies the discrete cluster that the latent attribute representation z_l is sampled from. If the reviewer has any specific comments about how this wording is confusing or suggestions for how it can be clarified, we would be more than happy to incorporate them.
> > Based on this notation system, we use y_o (the observed class) to denote the categorical variable that represents speaker IDs. The cardinality of y_o is the number of speakers in the training set. We will update the text in the paper to make this more explicit.
> > - Regarding computation of these two variables, since the observed class y_o is an observed variable (i.e. the speaker ID of an utterance), we do not need to compute it. In contrast, the latent class y_l is a latent variable, and we compute the posterior of y_l given an utterance using Eq. (2), whose detailed derivation can be found in Appendix A.1.
> > - Regarding initialization, each of the K possible values of y_l is associated with a diagonal-covariance Gaussian distribution which models the conditional distribution of z_l for that mixture component. Let D be the dimensionality of z_l. We implement this using two K-by-D lookup tables, one to store the mean of each mixture component, and the other for the corresponding log variance. Both the mean and the log variance table are trainable. The variances are initialized to a constant value listed in Table 6. Initial values of the mean is set with Xavier Initialization (Glorot and Bengio, 2010). For the extension proposed in Section 2.3, each value of y_o is also associated with a diagonal-covariance Gaussian that models the conditional distribution of z_o for a speaker. Likewise, the mean table is initialized using Xavier initialization and the initial value of the variance is set to a constant listed in Table 6. We will update the text to clarify this.
> > - Regarding setting, we are not certain if the reviewer meant “how to generate samples conditioning on some values of y_l/y_o.” Please clarify if we misinterpreted. If our interpretation is correct, for the proposed model shown in Figure 1 (left), we sample z_l from the conditional distribution of the y_l-th mixture component, which is diagonal-covariance Gaussian whose mean and log variance are specified in two lookup tables. For the extended proposed model shown in Figure 1 (right), we sample z_l in the same way as mentioned above, and similarly sample z_o from the conditional distribution of the y_o-th mixture components, which is also diagonal-covariance Gaussian whose mean and log variance are specified in another two lookup tables.

---

> > > ### Author Response · Authors · 2018-11-14
> > > **Response to AnonReviewer1 (Part 3)**
> > >
> > >
> > > Re:
> > > Why both negative results are in the appendix?
> > >
> > > Ans:
> > > - We are not certain which results the reviewer was referring to here. There are results in Table 8 and Table 9 where the proposed model does not have the best performance. If this comment refers to something else, please clarify so we can better address the concern.
> > > - First of all, we would like to point out that 1) These results are not really negative as explained in the accompanying text, and 2) We were not trying to hide anything. These results were not put it in the main text simply because we felt that they were less critical to validating the main contributions of the paper compared to those in the main text, and we did not have sufficient space to describe extensive experiments.
> > > - Here we will discuss each experiment in greater detail.
> > > -- For the speaker similarity test in Table 9 in Appendix G.2, our proposed model is compared with ground truth or other models on three sets: seen clean speakers (SC), seen noisy speakers (SN), and unseen clean speakers (UC).
> > > --- On the UC set, we compare with d-vector systems (Jia et al., 2018), which is a Tacotron-based system that utilizes a separately trained speaker verification system to extract speaker embeddings. To draw an analogy to our framework, the d-vector model can be regarded as training the q(z_o | X) module separately using a speaker verification task. In Table 9, we compare the proposed model to two d-vector systems, one of which utilizes a speaker verification system trained on the same training set as our proposed model, which contains ~1k speakers, and the other utilizes a speaker verification system trained on a much larger set that contains ~18k speakers. Our proposed model outperforms the former significantly (2.79 vs 2.23), which is the system that uses exactly the same amount of resources as our proposed model and hence is a fair comparison. The proposed model is worse than the latter d-vector-based model: d-vector (large). However we emphasize that this is not a fair comparison, as that d-vector system utilizes a speaker representation extractor *trained on 20 times more speakers than the proposed model*. Furthermore, we point out that our proposed model is complementary to d-vector systems. Incorporating the high quality speaker transfer from the d-vector model with the strong controllability in the GMVAE is something we intend to pursue in future work.
> > > --- On the SN set, we explain in text that “similarity of the acoustic conditions between the paired utterances biased the speaker similarity ratings.” In other words, even when both utterances come from the same speaker, we found that the raters still tended to assign a lower similarity score to the pair with different acoustic conditions than the pair with the matching acoustic conditions. We verified this phenomenon by showing that the human rated speaker similarity score for a speaker that has recordings with significant variation in acoustic conditions (Ground truth w/ channel variation) is much lower (3.30), compared to the ground truth similarity score of the clean speakers (4.33) reported in Jia et al, 2018. The baseline without denoising achieves the highest MOS score (3.83), because it reproduces the noise of the noisy speakers, while the baseline+denoise model and our proposed model have lower MOS scores because the acoustic conditions are changed comparing to the reference noisy utterances. This result implies that this subjective similarity test is not reliable in the presence of acoustic condition variation, and requires additional work to design speaker similarity test that is unbiased to these nuisance factors.
> > > -- The parallel style transfer experiment in Appendix F.1 evaluates the ability of the model to reconstruct the reference utterance frame-by-frame, which is highly correlated with the latent space capacity as noted in Skerry-Ryan et al, 2018. We showed in the table that increasing the latent space dimensionality of z_l in our proposed model from 16 to 32 greatly reduces the MCD and FFE gap to the GST model which performs best, and explained in text that the GST model still has more degrees of freedom in the latent space and a larger number of parameters than our proposed model. We can expect the model to outperform GST on these two metrics if we further increase the dimensionality of z_l; however, this would likely reduce interpretability and generalization of the latent attribute control.

---

> > > > ### Author Response · Authors · 2018-11-14
> > > > **Response to AnonReviewer1 (Part 4)**
> > > >
> > > >
> > > > Re:
> > > > What are different modes using the proposed model?
> > > >
> > > > Ans:
> > > > - We are not sure if the reviewer meant “what are the attributes presented in the generated samples when conditioning on the mode of each mixture component?” If we are interpreting it wrong, we would like to ask the reviewer for clarification so we can address the question correctly.
> > > > - The mode (mean) of each mixture component presents the average latent attributes when conditioned for generation. We demonstrated in Figure 3 examples of conditioning on two different modes for generation.
> > > >
> > > >
> > > > Re:
> > > > For example a reliable numerical evidence is needed on page 4 following "We also found...", page 5 following "We discovered....", page 6 following "Figure 7(b)...".
> > > >
> > > > Ans:
> > > > We are looking into metrics to quantitatively evaluate the effect of varying different latent dimensions on average pitch and speech duration now, and will update this thread once the results are available.  But see our response above for caveats about the reliability of this approach.
> > > >
> > > >
> > > > References:
> > > > Kingma, Diederik P., and Prafulla Dhariwal. "Glow: Generative flow with invertible 1x1 convolutions." arXiv preprint arXiv:1807.03039 (2018).
> > > > Wang, Yuxuan, et al. "Style Tokens: Unsupervised Style Modeling, Control and Transfer in End-to-End Speech Synthesis." in International Conference on Machine Learning (ICML), 2018.
> > > > Akuzawa, Kei, Yusuke Iwasawa, and Yutaka Matsuo. "Expressive Speech Synthesis via Modeling Expressions with Variational Autoencoder." in Interspeech, 2018.
> > > > Glorot, Xavier, and Yoshua Bengio. "Understanding the difficulty of training deep feedforward neural networks." in International Conference on Artificial Intelligence and Statistics (AISTATS), 2010.
> > > > Jia, Ye, et al. "Transfer Learning from Speaker Verification to Multispeaker Text-To-Speech Synthesis." arXiv preprint arXiv:1806.04558 (2018).
> > > > Skerry-Ryan, R. J., et al. "Towards End-to-End Prosody Transfer for Expressive Speech Synthesis with Tacotron." in International Conference on Machine Learning (ICML), 2018.

---

### Official Review · AnonReviewer3 · 2018-11-03
**A good work**

**Rating:** 6
**Confidence:** 5

**Review:**

This paper proposes a two layer latent variable model to obtain disentangled latent representation, thus facilitates fine-grained control over various attributes including noise level, speaker rate etc.

Detailed comments:

i) This work is closely related to Akuzawa et al. (2018). The difference is not properly discussed.

ii) In the abstract, “end-to-end text-to-speech” is an unfortunate claim, because the proposed system has two separately trained component: 1) a text to mel-spectrogram model based on Tacotron and, 2) a WaveRNN for waveform synthesis.  In ASR, it’s absolutely fine to claim a spectrogram to text model as a end-to-end system, because the wave to spectrogram step is trivial.  In TTS, waveform synthesis is a very crucial step and largely determines the final naturalness results.

iii) In experiment, one need include the MOS score of ground truth for comparison or debiasing.

iv) Did you try different values of D other than 16? When you check the meaning of different dimensions, how many dimensions are meaningful? How many are meaningless, or even just dummy?

In summary,
pros:
- A good work with impressive results.
cons:
- Related work need to be properly discussed.
- Doesn’t include the MOS of ground truth.
- Moderate novelty.

---

> ### Author Response · Authors · 2018-11-14
> **Response to AnonReviewer3 (Part 1)**
>
> We thank the reviewer for the great feedback. Below are the itemized responses regarding each comment. We will also incorporate these into the revised version.
>
> Due to the "max 5000 characters" limit, we break down our response into multiple comments.
>
>
> Re:
> This work is closely related to Akuzawa et al. (2018). The difference is not properly discussed.
>
> Ans:
> - There are two major differences from the latent space modeling perspectives, both of which have large effects on the interpretability of the latent representation and on the ability to control individual attributes. In addition, there is a difference in terms of the adopted neural network architecture, which we think is of less importance.  We will update the text to better describe these differences.
> -- First, we model latent attributes using a *mixture distribution*, which allows automatic discovery of latent attribute clusters. This design makes the resulting latent space much more interpretable, as data-driven clustering leads to meaningful structure. Specifically, we can 1) quickly analyze clusters to get a sense of what they correspond to (similar to GST), as in multispeaker experiments (Section 4.1), and 2) easily analyze inter-cluster variance to identify most distinctive attributes, which e.g. allows us to easily identify the noise dimension (Section 4.2.2). We demonstrate how identifying this structure can be easily used to train a model on noisy found-data that can consistently synthesize clean speech, which has not been done before to the best of our knowledge. Such interpretability cannot be achieved using an isotropic Gaussian prior as in Akuzawa et al., (2018).
> -- Second, we describe an extension of the proposed framework to additionally model speaker attributes with a second mixture distribution, where each speaker corresponds to one mixture component. This formulation learns disentangled speaker and latent attribute representations, which can be applied to one-shot learning to imitating the voice of speakers previously unseen during training (Section 4.4 and Appendix G.2). This functionality was not provided in Akuzawa et al. (2018) either. We will include this discussion in the next version.
> -- Third, regarding the neural network architecture, while Akuzawa et al. (2018) uses the VoiceLoop (Taigman et al., 2018) architecture for the synthesizer, our proposed model is based on Tacotron 2 (Shen et al., 2018). However, we would like to emphasize that such a choice is not the major difference in terms of the ability to control latent attributes, and our approach can also be applied to the VoiceLoop architecture.
> - Asides from the three differences mentioned above, we would also like to point out that our model synthesizes speech with higher quality than the samples presented on the demo page from Akuzawa et al. (2018) (https://akuzeee.github.io/VAELoopDemo/).
>
>
> Re:
> ii) In the abstract, “end-to-end text-to-speech” is an unfortunate claim, because the proposed system has two separately trained component: 1) a text to mel-spectrogram model based on Tacotron and, 2) a WaveRNN for waveform synthesis. In ASR, it’s absolutely fine to claim a spectrogram to text model as a end-to-end system, because the wave to spectrogram step is trivial. In TTS, waveform synthesis is a very crucial step and largely determines the final naturalness results.
>
> Ans:
> We agree with the reviewer and will delete the term “end-to-end” to reflect that vocoding from Mel-spectrogram is done with a separate WaveRNN module. In addition, we would like to restate that the focus of this paper is the ability to control latent attributes, which is achieved by the GMVAE-Tacotron module, and having the WaveRNN vocoder only improves the audio fidelity, similar to the observation made in GST (Wang et al., 2018). For example, we observed that the WADA-SNR metric has no difference when using Griffin-Lim or WaveRNN for vocoding.
>
>
> Re:
> iii) In experiment, one need include the MOS score of ground truth for comparison or debiasing.
>
> Ans:
> We are now evaluating the MOS score of ground truth, and will reply once the evaluation is finished.
>
> EDIT: We have obtained the ground truth MOS scores. Please see the comment below.

---

> > ### Author Response · Authors · 2018-11-14
> > **Response to AnonReviewer3 (Part 2)**
> >
> >
> > Re:
> > iv) Did you try different values of D other than 16? When you check the meaning of different dimensions, how many dimensions are meaningful? How many are meaningless, or even just dummy?
> >
> > Ans:
> > - As shown in Table 8, we found that increasing the dimensionality of z from 16 to 32 improves reconstruction quality; however, it also increases the difficulty of interpreting each dimension. On the other hand, reducing the dimensionality too much would result in insufficient modeling capacity for latent attributes, however we have not carefully explored this lower bound.
> > - Empirically, we found 16-dimensional z_l to be appropriate for capturing the salient attributes one would like to control in the four datasets we experimented with. When checking the meaning of each dimensions, we find the majority of the dimensions to be interpretable, and the number of dummy dimensions which did not seem to affect the model output varied across datasets, as each of them inherently has variation across a different number of unlabeled attributes. For example, the model trained on the multi-speaker English corpus (Section 4.1) has are four dummy dimensions of z_l that do not affect the output. On the other hand, for the model trained on the crowd-sourced audio book corpus (Section 4.4), which contains considerably more variation in style and prosody, we only found one dummy dimension of z_l.
> >
> >
> > Re:
> > Moderate Novelty
> >
> > Ans:
> > - With respect to the novelty of this paper, although we build off of strong previous work, we believe the proposed model is a significant advance compared to the existing state of the art in controllable neural TTS. We propose a novel and more principled probabilistic hierarchical generative model which significantly improves 1) disentangled attribute control compared to e.g. the GST model of Wang et al. (2018), and 2) interpretability as well as quality compared to e.g. Akuzawa et al. (2018). Another advantage compared to the GST model is that the proposed model has a straightforward sampling procedure, making it easy to generate varied samples for the same text.
> > - Using two mixture distributions to factor the latent encoding and separately model speaker attributes and latent attributes is another novel aspect of this work. This formulation enables the inference model to disentangle speaker and latent attribute representations by encoding them into separate latent variables, which can be applied to condition the generation on the speaker attribute and latent attribute inferred from different reference utterances. This functionality was not provided in any previous work.
> > - In addition, this work is the first that we know of to successfully train a high-quality controllable text-to-speech system on real found data containing significant variation in audio quality, acoustic conditions, as well as prosody and style. Previous results on this dataset focused only on speaker modeling (Ping, et al 2017, Nachmani, et al 2018, Arik, et al 2018, Jia, et al 2018), and did not address explicit control of prosody and background noise. Our proposed model is capable of inferring the speaker attribute representation from a noisy utterance of a previously unseen speaker, and using it to synthesize high-quality clean speech that approximates the voice of that unseen speaker. To the best of our knowledge, no other work has accomplished this.
> > - We will edit the abstract and introduction to better highlight the novelty of our contribution in the next revision of the text.
> >
> >
> > References:
> > Akuzawa, Kei, Yusuke Iwasawa, and Yutaka Matsuo. "Expressive Speech Synthesis via Modeling Expressions with Variational Autoencoder." in Interspeech, 2018.
> > Taigman, Yaniv, et al. "VoiceLoop: Voice Fitting and Synthesis via a Phonological Loop." in International Conference on Learning Representations (ICLR), 2018.
> > Shen, Jonathan, et al. "Natural TTS Synthesis by Conditioning WaveNet on Mel Spectrogram Predictions." International Conference on Acoustics, Speech and Signal Processing (ICASSP), 2018.
> > Wang, Yuxuan, et al. "Style Tokens: Unsupervised Style Modeling, Control and Transfer in End-to-End Speech Synthesis." in International Conference on Machine Learning (ICML), 2018.
> > Ping, Wei, et al. "Deep voice 3: 2000-speaker neural text-to-speech." arXiv preprint arXiv:1710.07654 (2017).
> > Nachmani, Eliya, et al. "Fitting New Speakers Based on a Short Untranscribed Sample." arXiv preprint arXiv:1802.06984 (2018).
> > Arik, Sercan O., et al. "Neural Voice Cloning with a Few Samples." arXiv preprint arXiv:1802.06006 (2018).

---

> > ### Author Response · Authors · 2018-11-15
> > **MOS of ground truth**
> >
> > Below are the MOS scores of the ground truth, which we will add to the corresponding tables in the paper text:
> > - Multi-speaker English corpus (top-line for Table 1): 4.48 ± 0.04
> > - Single-speaker audiobook corpus (Table 2): 4.67 ± 0.04
> > - Crowd-sourced audiobook corpus (Table 5):
> >   - seen clean (SC): 4.60 ± 0.07
> >   - seen noisy (SN): 4.45 ± 0.08
> >   - unseen clean (UC): 4.54 ± 0.08
> >   - unseen noisy (UN): 4.34 ± 0.08
> > Note that the ground truth MOS score of the multi-speaker English corpus serves as the very top-line for models trained on Noisy multi-speaker English corpus (Section 4.2), because those audio samples are clean and represent the best case scenario compared to the data used for training models shown in Table 1, which had noised mixed in.
> > The proposed model achieved comparable MOS to the ground truth on the single-speaker audiobook corpus. On the multi-speaker audiobook corpus, the MOS scores of the proposed model are 0.14 - 0.42 behind those of the ground truth. For the model trained on the noisy multi-speaker English corpus, the MOS score is 0.23 lower than the very top-line.

---

### Official Review · AnonReviewer2 · 2018-11-06
**A structure to the z-space to create predictabilily in the features of the generated speech is implemented  with good results**

**Rating:** 8
**Confidence:** 4

**Review:**

The authors describe the conditioned GAN model to generate speaker conditioned Mel spectra. They augment the z-space corresponding to the identification with latent variables that allow a richer set of produced audio. In a way this is like a partially conditioned model that has "extra" degrees of freedom. It looks that the "latent" variables are just concaneted to the "original" set of z-values (altough with particular conditions to maximize independence). The conditioning of the z-space has originality in it and may provide interesting to the audience. Ultimately one coud think about z-space direction being totally mapped to specific features of the produced signal.

Also, I am curious to know how the Mel spectra are used to produce the actual sound wave - as the phase information is not present if utilizing only the spectral amplitude. Very often this leads to suboptimal generation, and the remedy is to use the time domain like in ( https://arxiv.org/ftp/arxiv/papers/1810/1810.05319.pdf).  However, in this case the audio samples show a pretty nice generation of sound.  However, it is not really end to end.

The manuscript has some curious decisios in its concepts - I do not see the architecture really hiearchial, nor end to end. I would prefer modifications on the paper that concentrate on the truly novel features.

The paper is clear, well written and done with high ambition, from data set utilization  to novel architetures to human quality panels. Results are good and interesting.

NEW:
The authors have addressed the concerns I had with the manuscript.

---

> ### Author Response · Authors · 2018-11-13
> **Response to AnonReviewer2 (Part 1)**
>
> We thank the reviewer for the great feedback. Below are the itemized responses regarding each comment. We will also incorporate these into the revised version.
>
> Due to the "max 5000 characters" limit, we break down our response into multiple comments.
>
>
> Re:
> The authors describe the conditioned GAN model to generate speaker conditioned Mel spectra. They augment the z-space corresponding to the identification with latent variables that allow a richer set of produced audio. In a way this is like a partially conditioned model that has "extra" degrees of freedom. It looks that the "latent" variables are just concaneted to the "original" set of z-values (altough with particular conditions to maximize independence). The conditioning of the z-space has originality in it and may provide interesting to the audience. Ultimately one coud think about z-space direction being totally mapped to specific features of the produced signal.
>
> Ans:
> - The proposed model is not a conditional GAN, but is in fact a conditional VAE which uses a Gaussian mixture prior for latent attribute representations (z_l), and a Gaussian mixture prior for speaker attribute representations (z_o) where each speaker corresponds to one mixture component. While the speaker identity (y_o) is given, the latent attribute representations and speaker attribute representations (z_l and z_o) are both continuous latent variables, the posteriors of which are estimated from an utterance by neural networks, denoted in Figure 8 as latent encoder and observed encoder, respectively.
> - We are not sure if the reviewer is referring to z_o as the “original set of z-values” and z_l as the “latent variables.” As mentioned above, during training, both z_o and z_l used for generating the target speech X are latent variables inferred directly from the spectrogram, as shown in Eq. (4). This is different from the typical speaker embedding-based method, which uses a fixed embedding learned for each speaker stored in an embedding table. From the perspective of modeling speakers, our models has the advantage that it is capable of inferring the z_o for an unseen speaker and imitate the voice of that unseen speaker, while the speaker embedding method cannot.
>
>
> Re:
> Also, I am curious to know how the Mel spectra are used to produce the actual sound wave - as the phase information is not present if utilizing only the spectral amplitude…. However, in this case the audio samples show a pretty nice generation of sound. However, it is not really end to end.
>
> Ans:
> - To synthesize waveforms from Mel spectrograms, we closely follow the Tacotron 2 (Shen et al, 2018) framework, which trains a neural network-based vocoder to predict waveforms conditioning on the Mel-spectrograms generated by GMVAE-Tacotron. In this work, we use WaveRNN (Kalchbrenner et al, 2018) as the neural vocoder, as described in Section 2.4. We will update the text to further clarify this.
> - We agree with the reviewer about the wording and will delete the term “end-to-end” to reflect the use of neural vocoder. In addition, we would like to restate that the focus of this paper is the ability to control latent attributes, which is achieved by the GMVAE-Tacotron module, and having the WaveRNN vocoder only improves the audio fidelity, similar to the observation made in GST (Wang et al., 2018). For example, we observed that the WADA-SNR results have no difference when using Griffin-Lim or WaveRNN for vocoding.

---

> > ### Author Response · Authors · 2018-11-14
> > **Response to AnonReviewer2 (Part 2)**
> >
> > Re:
> > The manuscript has some curious decisios in its concepts - I do not see the architecture really hiearchial, nor end to end. I would prefer modifications on the paper that concentrate on the truly novel features.
> >
> > Ans:
> > - As illustrated in Figure 1, the VAE prior distribution is a hierarchical model -- i.e. a Gaussian mixture model, where one first samples a mixture component y_l, and then samples a continuous vector z_l conditioning on that mixture component. By introducing such a structured prior, our model provides better interpretability compared to the model which uses an isotropic Gaussian prior for latent attributes (Akuzawa et al, 2018). For example, our model can automatically discover clusters of latent attributes (Section 4.1), and we easily analyze inter-cluster variance with linear discriminative analysis to identify the most distinctive attributes, e.g. as in identifying the noise dimension (Section 4.2.2). Designing such a hierarchical structure to model latent attributes for interpretability is our main contribution, which we feel is clear from the text. The sequence-to-sequence speech synthesizer (bottom block in Figure 8) itself is based on standard architecture in the literature (Wang et al., 2017 and Shen et al., 2018), and is indeed not especially hierarchical.
> > - Regarding “end-to-end”, as mentioned in the response to the previous question, we agree with the reviewer and will delete the term from the text.
> >
> >
> > References:
> > Shen, Jonathan, et al. "Natural TTS Synthesis by Conditioning WaveNet on Mel Spectrogram Predictions." International Conference on Acoustics, Speech and Signal Processing (ICASSP), 2018.
> > Kalchbrenner, Nal, et al. "Efficient Neural Audio Synthesis."  in International Conference on Machine Learning (ICML), 2018.
> > Wang, Yuxuan, et al. "Style Tokens: Unsupervised Style Modeling, Control and Transfer in End-to-End Speech Synthesis." in International Conference on Machine Learning (ICML), 2018.
> > Akuzawa, Kei, Yusuke Iwasawa, and Yutaka Matsuo. "Expressive Speech Synthesis via Modeling Expressions with Variational Autoencoder." in Interspeech, 2018.
> > Wang, Yuxuan, et al. "Tacotron: Towards end-to-end speech synthesis." in Interspeech, 2017.

---

### Public Comment · (anonymous) · 2018-11-14
**Great result but some important details are missing**

Authors describe a way to learn disentagled inner representations for prosody using variational autoencoder approach. This allows for fine-grained control of synthesized speech parameters such as tempo, accent, etc.

Presented results are stunning, the writing is mostly clear. My concerns are:

1. Quote (B.1): "To condition the output on additional attribute representations, the decoder is extended to consume z_l and z_o (or y_o) after passing them through a stack of two uni-directional LSTM layers with 1024 units. The output from the stacked LSTM is concatenated with the decoder input, and linearly projected to predict the mel spectrum of the current frame, as well as an end-of-sentence token."
It follows from the quote that attibute representations are passed through recurrent layers, while the next subsection (B.2) describes a mean pooling component for summarizing outputs across time in latent and observed encoders, so encoders' outputs are fixed-dimension vectors. How would one pass a fixed vector through recurrent layers? Moreover, why output of these recurrent layers is concatenated with decoder input to predict next frame? Should not it be concatenated with decoder output? What exactly word 'decoder' signifies in this section?

A more careful and detailed description of modifications to Tacotron 2 architecture is desirable.

2. Learning a good latent disentangled representation using VAE is known to be not an easy task (https://openreview.net/references/pdf?id=Sy2fzU9gl)
Actually, in the last paragraph of 'Related work' section the first referenced paper [Dilokthanakul et al., 2016] contains a discussion. Some quotes from section 3.4 ('The Over-regularisation Problem'):
1) 'The possible overpowering effect of the regularisation term on VAE training has been described numerous times in the VAE literature (<references>)'
2) 'As we show in the experimental section below, this problem of over-regularisation is also prevalent in the assignment of the GMVAE clusters and manifests itself in large degenerate clusters'.
In this same section Dilokthanakul et al. briefly (and with references) describe possible approaches (such as KL-term annealing) to mitigating these problems. One of the approaches is discussed in section 4.1.

Unfortunately, the paper does not touch this topic. It is unclear whether authors of GMVAE-Tacotron follow any of these approaches or succeed entirely without doing so.

3. Number of Monte Carlo samples for estimating the objective is set to 1 with no further discussion.

---

> ### Author Response · Authors · 2018-11-16
> **Response to anonymous reviewer (Part 1)**
>
> We thank the reviewer for the great feedback. Below are the itemized responses regarding each comment. We will also incorporate these into the revised version.
>
> Due to the "max 5000 characters" limit, we break down our response into multiple comments.
>
>
> Re:
> It follows from the quote that attibute representations are passed through recurrent layers … How would one pass a fixed vector through recurrent layers? Moreover, why output of these recurrent layers is concatenated with decoder input to predict next frame? Should not it be concatenated with decoder output? What exactly word 'decoder' signifies in this section?
>
> Ans:
> - We thank the reviewer for pointing out the confusion about “decoder”. The definition of “decoder” in this work follows the convention in auto-regressive sequence-to-sequence modeling literature (Sutskever et al., 2014, Bahdanau et al., 2015), which parameterizes $p(X | z_l, z_o, text_encodings)$. The text_encoding is computed using an encoder network that takes text ($y_t$) as input. The conditional distribution $p( X | z_l, z_o, y_t)$ which generates the output spectrogram (usually termed “decoder” in the variational autoencoder literature (Kingma et al., 2014)) is in fact parameterized by both the text encoder and the auto-regressive decoder as $Decoder(z_l, z_o, TextEncoder(y_t) )$ in our work.
> - The quoted text describes the computation involved at each step of the auto-regressive decoder. In our model, the decoder input at each timestep is a concatenated vector composed of four vectors: $z_l$, $z_o$ (or $y_o$), bottlenecked previous output frame, and attention-aggregated text encoding. This decoder input is passed to two layers of LSTM units. The output from the final-layer LSTM unit is concatenated with the decoder input, and passed to a linear layer to predict the mel spectrum of the current frame and the binary end-of-sentence token.
> - At each decoder step, the same $z_l$ and $z_o$ are used for concatenating with the other two vectors. In other words, $z_l$ and $z_o$ are used for global conditioning across all frames in the output mel-spectrum sequence.
>
>
> Re:
> In the “Related work” section the first referenced paper [Dilokthanakul et al., 2016] contains a discussion. Some quotes from section 3.4 ('The Over-regularisation Problem'):
> 1) 'The possible overpowering effect of the regularisation term on VAE training has been described numerous times in the VAE literature (<references>)'
> 2) 'As we show in the experimental section below, this problem of over-regularisation is also prevalent in the assignment of the GMVAE clusters and manifests itself in large degenerate clusters'.
> In this same section Dilokthanakul et al. briefly (and with references) describe possible approaches (such as KL-term annealing) to mitigating these problems. One of the approaches is discussed in section 4.1.
> Unfortunately, the paper does not touch this topic. It is unclear whether authors of GMVAE-Tacotron follow any of these approaches or succeed entirely without doing so.
>
> Ans:
> - There are two latent variables in our graphical model as shown in Figure 1(left): latent attribute class $y_l$ (discrete) and latent attribute representation $z_l$ (continuous). We will discuss the posterior-collapse problem for each of them separately.
>
> - The continuous latent variable $z_l$ is used to directly condition the generation of $X$, along with two other observed variables, $y_o$ and $y_t$. In our experiments, we observed that the latent variable $z_l$ is always used (i.e., the KL-divergence of $z_l$ never drops to zero), without applying any tricks such as KL-annealing. Previous studies report posterior-collapse of directly conditioned latent variables when using strong models (e.g. auto-regressive model) to parameterize the conditional distribution of text (Bowman et al., 2015, Kim et al., 2018). This phenomenon arises from the competition between (1) increasing reconstruction performance by utilizing information provided by the latent variable and (2) decreasing the KL-divergence by having an uninformative latent variable. Auto-regressive models are more likely to converge to the second case during training because the improvement in reconstruction from utilizing the latent variable can be smaller than the increase in KL-divergence. However, this does not always happen, because the amount of improvement resulted from utilizing the information provided by the latent variable depends on the type of data. The reason that the posterior-collapse does not occur in our experiments is likely a consequence of the complexity of the speech sequence distribution, compared to text, such that even when we have an auto-regressive model, reconstruction performance can still be improved significantly by utilizing the information from $z_l$, and such improvement overpowers the increase in KL-divergence.

---

> > ### Author Response · Authors · 2018-11-16
> > **Response to anonymous reviewer (Part 2)**
> >
> >
> > Ans (continued):
> > - The discrete latent variable $y_l$ indexes the mixture components in the space of latent attribute representation $z_l$. We did not observe the phenomenon of degenerate clusters mentioned in Dilokthanakul et al. (2016) when training our model using the hyperparameters listed in Table 6. Below we identify the difference between our GMVAE and that in Dilokthanakul et al. (2016), which we will refer to as Dil-GMVAE, and explain why our formulation is less likely to have the posterior-collapse issue.
> >
> > - *Difference between GMVAE and Dil-GMVAE*: We first unify the notation system as it is different in the two papers. Let $X$ be the observed data we want to generate (speech), let $y$ be the discrete latent variable that denotes the mixture components, and let $z$ be the continuous latent variable that is conditionally sampled based on $y$ to directly condition the generation of $X$.
> > In our model, the conditional distribution $p(z | y)$ is a diagonal-covariance Gaussian, $N(z; \mu_y, \sigma^2_y)$, parameterized by a mean and a covariance vector. In contrast, in Dil-GMVAE, the conditional distribution of $z$ given $y$ is much more flexible, because it is parameterized using neural networks as $p(z | y) = \int_\epsilon N( z; NN_{\mu}(y, \epsilon), NN_{\sigma^2}(y, \epsilon) ) N(\epsilon; 0, I) d\epsilon$, where $NN_{\mu}$ and $NN_{\sigma^2}$ are neural networks that take $y$ and an auxiliary noise variable $\epsilon$ as input to predict the mean and variance of $p(z | y, \epsilon)$. The conditional distribution of each component in Dil-GMVAE can be seen as a mixture of infinitely many diagonal-covariance Gaussian distributions, which can model much more complex distributions, as shown in Figure 2(d) in Dilokthanakul et al. (2016). That is to say, Dil-GMVAE can be regarded as having a much stronger stochastic decoder that maps y to z compared with our GMVAE.
> >
> > - *why our formulation is less likely to have the posterior-collapse issue*: Suppose the decoder that maps $z$ to $X$ can benefit from having a very complex, non-Gaussian marginal distribution of $z$, denoted as $p^*(z)$. To obtain such a marginal distribution of $z$ in Dil-GMVAE, the $y$->$z$ decoder can choose between (1) having the same $p(z | y) = p^*(z)$ for all $y$, or (2) having $p(z | y)$ model a different distribution for each $y$, and $\sum_y p(z | y) p(y) = p^*(z)$. As noted in Dilokthanakul et al. (2016), the KL-divergence term for $y$ in the ELBO prefers degenerate clusters that basically model the same distribution for each cluster. As a result, the first option would be preferred with respect to the ELBO objective compared to the second one, because it does not compromise the expressiveness of $z$ while minimizing the KL-divergence on $y$. In contrast, our GMVAE reduces to a single Gaussian when $p(z | y)$ is the same for all $y$, and hence there is a trade-off between the expressiveness of $p(z)$ and the KL-divergence on $y$.
> >
> > - In addition, we now explain the connection between posterior-collapse and hyperparameters of the conditional distribution $p(z | y)$ in our work. In our GMVAE model, posterior-collapse of $y$ is equivalent to having the same conditional mean and variance for each mixture component. In the ELBO derived from our model, there are two terms that are relevant to $p(z | y)$, which are (1) the expected KL-divergence on $z$: $E_{q(y | X)} [ KL(q(z | X) || p(z | y)) ]$ and (2) the KL-divergence on $y$: $KL(q(y | X) || p(y))$. The second term encourages a uniform posterior $q(y | X)$, which effectively pulls the conditional distribution for each component to be close to each other, and promotes posterior collapse. In contrast, the first term pulls each $p(z | y)$ to be close to $q(z | X)$ with a force proportional to the posterior of that component, $q(y | X)$. In one extreme, where the posterior $q(y | X)$ is close to uniform, each $p(z | y)$ is also pushed toward the same distribution, $q(z | X)$, which promotes posterior collapse. In the other extreme, where the posterior $q(y | X)$ is close to one-hot, only the conditional distribution of the assigned component $p(z | y)$ is pushed toward $q(y | X)$, and as long as different X are assigned to different components this term is anti-collapse. Therefore, we can see that the effect of the first term on posterior-collapse depends on the entropy of $q(y | X)$, which is controlled by the scale of the variance when the means are not collapsed. This variance is similar to the temperature parameter used in softmax: the smaller the variance is, the more spiky the posterior distribution over y is. This is why we set the initial variance of each component to a smaller value at the beginning of training, which helps avoid posterior collapse.

---

> > > ### Author Response · Authors · 2018-11-16
> > > **Response to anonymous reviewer (Part 3)**
> > >
> > > Re:
> > > Number of Monte Carlo samples for estimating the objective is set to 1 with no further discussion.
> > >
> > > Ans:
> > > This follows the common practice in the VAE literature (Kingma et al., 2014) when training a model with a batch size that is large enough.  We use a batch size of 256 in all experiments. To estimate the approximated posterior over latent attribute classes, q(y_l | X), using Eq (5), we experimented with using 1 or 10 samples of z_l drawn from q(z_l | X), and did not see much effect on training loss or disentanglement performance.
> > >
> > >
> > > References:
> > > Sutskever, Ilya, Oriol Vinyals, and Quoc V. Le. "Sequence to sequence learning with neural networks." Advances in neural information processing systems. 2014.
> > > Bahdanau, Dzmitry, Kyunghyun Cho, and Yoshua Bengio. "Neural machine translation by jointly learning to align and translate." arXiv preprint arXiv:1409.0473 (2014).
> > > Kingma, Diederik P., and Max Welling. "Auto-encoding variational bayes." ICLR (2014).
> > > Dilokthanakul, Nat, et al. "Deep unsupervised clustering with gaussian mixture variational autoencoders." arXiv preprint arXiv:1611.02648 (2016).
> > > Bowman, Samuel R., et al. "Generating sentences from a continuous space." arXiv preprint arXiv:1511.06349 (2015).
> > > Kim, Yoon, et al. "Semi-Amortized Variational Autoencoders." arXiv preprint arXiv:1802.02550 (2018).

---

> > > > ### Public Comment · (anonymous) · 2018-11-19
> > > > **Thanks and additional questions to Paper948 Authors**
> > > >
> > > > Thank you for such a thorough and valuable response!
> > > >
> > > > Concerning my first question: if I understood correctly you condition a submodule of the Tacotron decoder on output of the variational encoder. The Tacotron GST paper [1] contains a brief discussion of condition sites:
> > > > "We experimented with different combinations of conditioning sites, and found that replicating the style embedding and simply adding it to every text encoder state performed the best."
> > > >
> > > > Would you please elaborate on your choice?
> > > >
> > > > Also there's a couple of minor misprints:
> > > > 1) In Appendix C, you denote variance with a $\sigma$, which is usually used for standard deviation. Did you mean $\sigma^2$?
> > > > 2) In Appendix A.1, there is an equality sign between eq. 11 and eq. 12, despite absence of a denominator present in eq. 7. It is obvious that values could be easily renormalized to probabilities since the distribution is discrete, but still, an equality sign might not be the most appropriate.
> > > >
> > > > [1] Wang et al., "Style Tokens: Unsupervised Style Modeling, Control and Transfer in
> > > > End-to-End Speech Synthesis"

---

> > > > > ### Author Response · Authors · 2018-11-21
> > > > > **Re: "Thanks and additional questions to Paper948 Authors"**
> > > > >
> > > > > We thank the reviewer for the comment. Below are our itemized responses.
> > > > >
> > > > > Re: Would you please elaborate on your choice?
> > > > > Ans: We experimented with concatenating and adding, but did not observe significant difference based on subjective evaluation.
> > > > >
> > > > > Re: Also there's a couple of minor misprints...
> > > > > Ans: We thank the reviewer for pointing out these typos. These typos were fixed in the latest revision.

---

> > > > > > ### Public Comment · (anonymous) · 2018-11-22
> > > > > > **Re:**
> > > > > >
> > > > > > My apologies, it looks like wording was not clear enough. By 'choice' I meant choice of condition site: text embeddings vs. text encoder outputs vs. somewhere inside of the decoder. One could add or concatenate the variational encoder's outputs to either of those.
> > > > > >
> > > > > > Would you please elaborate on your choice of condition site?

---

> > > > > > > ### Author Response · Authors · 2018-11-26
> > > > > > > **Re:conditioning site**
> > > > > > >
> > > > > > > We thank the reviewer for clarification. The encoder submodule in the synthesizer aims to encode a sequence of text one-hot vectors into a sequence of text encodings, which are taken as input along with other attribute encodings by the decoder submodule. By conditioning the latent vectors at the encoder output, we modularize the encoder in the synthesizer to focus on only encoding the text, and designate the decoder to integrate information of text, latent attributes, and observed attributes from the three inputs in order to predict spectrograms. We did not experiment with injecting the information elsewhere in the synthesizer.

---

### Author Response · Authors · 2018-11-21
**A new revision uploaded**

We thank all reviewers for the thoughtful comments and constructive suggestions, which were extremely helpful to us in improving the paper. We have uploaded a new revision to address these comments. Major changes are listed below:

- List novel contribution of this work in the introduction (Section 1).

- Expand discussion of related work.

- Added naturalness MOS scores of ground truth audio samples in every table for comparison.

- Added discussion about the dimensionality of latent space in Appendix C.

- Added discussion about the potential for posterior collapse in Appendix D.

- Added objective quantitative evaluations for disentangled attribute control for models trained on noisy multi-speaker English corpus (Appendix F.3) and single-speaker audiobook corpus (Appendix G.1).

- Fixed typo in the q(y_l | X) derivation (Appendix A.1).

- Added more details describing the experiment setup (Appendix C).

- Revised synthesizer neural network descriptions (Appendix B.1).

- Moved the speaker similarity evaluation for crowd-sourced audiobook corpus to the main text (Section 4.4.2).

## EDIT (2018-11-26)
- Moved objective quantitative noise-level control evaluations from Appendix F.3 to Section 4.2.2, and changed the tables to plots.

---

### Public Comment · (anonymous) · 2018-12-04
**Details about latent encoders vague**

Great papers but the details in regard to the latent encoders were a bit vague.

"For each encoder, a mel spectrogram is first passed through two convolutional layers, which contains
512 filters with shape 3 × 1. The output of these convolutional layers is then fed to a stack of
two bidirectional LSTM layers with 256 cells at each direction. A mean pooling layer is used
to summarize the LSTM outputs across time, followed by a linear projection layer to predict the
posterior mean and log variance."

1. Are the convolutional layers 2D convolutions?
2. What are the paddings and strides for the convolutions?
3. If the convolutional layers are 2D, were the mel channels (80) and the filter channels (512) collapsed together for the input of the LSTM?
4. Mean pooling layer details are vague and unrepeatable.

Thank you.

---

> ### Author Response · Authors · 2018-12-12
> **Re: Details about latent encoders**
>
> We thank you for your interest. Below are the itemized responses to your questions.
>
> 1. We treat a mel spectrogram as a T-by-1 image with 80 channels (dimensionality of a mel spectrum), where T corresponds to the number of frames. Convolution is therefore effectively one-dimensional across time.
> 2. We use stride=1 and zero-paddings such that the width and height remain the same after each convolution layer.
> 3. We did not add a skip connection from the input to the bidirectional LSTM (BLSTM) input -- i.e., the input to the first layer BLSTM is the output from the last layer convolutional layer, which has depth of 512 at each time step.
> 4. The mean pooling layer is quite simple. The last layer of BLSTM outputs one 512-dimensional vector at each time step, i.e. the output for the full sequence is a T x 512 matrix containing T frames. The mean pooling layer averages these vectors (per-dimension) across time to produce a single 512-dimensional vector summary of the spectrogram.  We hope this clarifies things.

---

### Meta-Review · Area_Chair1 · 2018-12-15
**Paper attacks significant problem**

**Confidence:** 3
**Recommendation:** Accept (Poster)

**Metareview:**

This is an ambitious paper tackling the important and timely problem of controlling non-annotated attributes in generated speech.

The reviewers had mixed opinions about the results. R1 asks for more convincing exposition of results but, nevertheless, acknowledging that it is difficult to evaluate TTS systems systematically. Besides, R2 and R3 find the results good.

Judging from the reviews and previous work, this paper does not seem to be very novel, although it certainly has intriguing new elements. Furthermore, it constitutes a mature piece of work.